# A Representational Model of Grid Cells' Path Integration Based on Matrix Lie Algebras

## Abstract

The grid cells in the mammalian medial entorhinal cortex exhibit striking hexagon firing patterns when the agent navigates in the open field. It is hypothesized that the grid cells are involved in path integration so that the agent is aware of its self-position by accumulating its self-motion. Assuming the grid cells form a vector representation of self-position, we elucidate a minimally simple recurrent model for grid cells' path integration based on two coupled matrix Lie algebras that underlie two coupled rotation systems that mirror the agent's self-motion: (1) When the agent moves along a certain direction, the vector is rotated by a generator matrix. (2) When the agent changes direction, the generator matrix is rotated by another generator matrix. Our experiments show that our model learns hexagonal grid response patterns that resemble the firing patterns observed from the grid cells in the brain. Furthermore, the learned model is capable of near exact path integration, and it is also capable of error correction. Our model is novel and simple, with explicit geometric and algebraic structures.

## 1 Introduction

Imagine walking in the darkness. Purely based on your sense of self-motion, you can gain a sense of self-position by integrating the self movement - a process often referred to as path integration (Darwin, 1873; Etienne & Jeffery, 2004; Hafting et al., 2005; Fiete et al., 2008; McNaughton et al., 2006). While the exact neural underpinning of path integration remains unclear, it has been hypothesized that the grid cells (Hafting et al., 2005; Fyhn et al., 2008; Yartsev et al., 2011; Killian et al., 2012; Jacobs et al., 2013; Doeller et al., 2010) in the mammalian medial entorhinal cortex (mEC) may be involved in this process (Gil et al., 2018; Ridler et al., 2019; Horner et al., 2016). The grid cells are so named because individual neurons exhibit striking firing patterns that form hexagonal grid patterns when the agent (such as a rat) navigates in a 2D open field (Fyhn et al., 2004; Hafting et al., 2005; Fuhs & Touretzky, 2006; Burak & Fiete, 2009; Sreenivasan & Fiete, 2011; Blair et al., 2007; Couey et al., 2013; de Almeida et al., 2009; Pastoll et al., 2013; Agmon & Burak, 2020). The grid cells interact with the place cells in the hippocampus (O'Keefe, 1979). Unlike a grid cell that fires at the vertices of a lattice, a place cell often fires at a single or a few locations.

The purpose of this paper is to understand how the grid cells may perform path integration (or "path integration calculations"). We propose a representational model in which the self-position is represented by the population activity vector formed by grid cells, and the self-motion is represented by the rotation of this vector. Specifically, our model consists of two coupled systems: (1) When the agent moves along a certain direction, the vector is rotated by a generator matrix of a Lie algebra. (2) When the agent changes movement direction, the generator matrix itself is rotated by yet another generator matrix of a different Lie algebra. Our numerical experiments demonstrate that our model learns hexagon grid patterns which share many properties of the grid cells in the rodent brain. Furthermore, the learned model is capable of near exact path integration, and it is also capable of error correction.

Our model is novel and simple, with explicit geometric and algebraic structures. The population activity vector formed by the grid cells rotates in the "mental" or neural space, monitoring the egocentric self-motion of the agent in the physical space. This model also connects naturally to the basis expansion model that decomposes the response maps of place cells as linear expansions of response maps of grid cells (Dordek et al., 2016; Sorscher et al., 2019). Overall, our model provides a

new conceptual framework to study the grid cell systems in the brain by considering the structure of the intrinsic symmetry (through Lie algebra) of the task which the path integration system is solving.

## 2 REPRESENTATIONAL MODEL FOR PATH INTEGRATION

Consider an agent navigating within a squared domain (theoretically the domain can be $\mathbb{R}^2$). Let $\boldsymbol{x} = (x_1, x_2)$ be the self-position of the agent in a 2D environment. At self-position $\boldsymbol{x}$, if the agent makes a displacement $\delta r$ along the direction $\theta \in [0, 2\pi]$, then the self-position is changed to $\boldsymbol{x} + \delta\boldsymbol{x}$, where $\delta\boldsymbol{x} = (\delta x_1, \delta x_2) = (\delta r \cos\theta, \delta r \sin\theta)$. In our model, we use a *polar* coordinate system (see figure 1a,b) by directly using $(\theta, \delta r)$, while only keeping $(\delta x_1, \delta x_2)$ implicit. $(\theta, \delta r)$ is the biologically plausible egocentric representation of self-motion.

We assume that the location $\boldsymbol{x}$ in the 2D environment is encoded by the response pattern of a population of $d$ neurons (e.g., $d = 200$), which correspond to a $d$-dimensional vector $\boldsymbol{v}(\boldsymbol{x}) = (v_i(\boldsymbol{x}), i = 1, ..., d)^\top$, with each element representing the firing rate of one neuron when the animal is at location $\boldsymbol{x}$. From the embedding point of view, essentially we embed the 2D domain in $\mathbb{R}^2$ as a 2D manifold in a higher dimensional space $\mathbb{R}^d$. Locally we embed the 2D local polar system centered at $\boldsymbol{x}$ (see figure 1a,b) into $\mathbb{R}^d$ so that it becomes a local system around $\boldsymbol{v}(\boldsymbol{x})$ (see figure 1c).

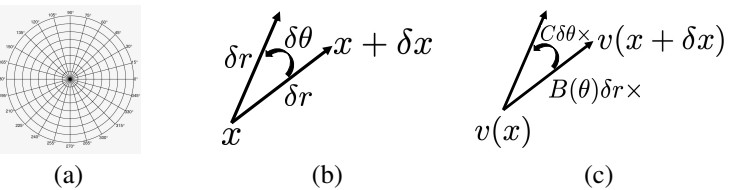

(a)  (b)  (c)

Figure 1: Illustration of the proposed representational model. (a) 2D local polar system centered at $\boldsymbol{x}$ for egocentric self-motion, to be embedded in $\mathbb{R}^d$. (b) 2D local displacement $\delta r$ and local change of direction $\delta\theta$. (c) Mirroring relations in (b). $\boldsymbol{x}$ is mirrored by $\boldsymbol{v}(\boldsymbol{x})$. Local displacement $\delta r$ from $\boldsymbol{x}$ along direction $\theta$ is mirrored by $\boldsymbol{B}(\theta)\delta r$ applied to $\boldsymbol{v}(\boldsymbol{x})$. Local change of direction $\delta\theta$ is mirrored by $\boldsymbol{C}\delta\theta$ applied to $\boldsymbol{B}(\theta)$.

### 2.1 THE PROPOSED REPRESENTATIONAL MODEL: COUPLING TWO ROTATION SYSTEMS

Assuming $\delta r$ to be infinitesimal, we propose the following model

$$\boldsymbol{v}(\boldsymbol{x} + \delta\boldsymbol{x}) = (\boldsymbol{I} + \boldsymbol{B}(\theta)\delta r)\boldsymbol{v}(\boldsymbol{x}) + o(\delta r), \tag{1}$$

which parameterizes a recurrent neural network (Hochreiter & Schmidhuber, 1997), where $\boldsymbol{I}$ is the identity matrix, and $\boldsymbol{B}(\theta)$ is a $d$-dimensional matrix depending on the direction $\theta$, which will need to be learned.

**Rotation**. We assume $\boldsymbol{B}(\theta) = -\boldsymbol{B}(\theta)^\top$, i.e., skew-symmetric, so that $\boldsymbol{I} + \boldsymbol{B}(\theta)\delta r$ is a rotation or orthogonal matrix, due to that $(\boldsymbol{I} + \boldsymbol{B}(\theta)\delta r)(\boldsymbol{I} + \boldsymbol{B}(\theta)\delta r)^\top = \boldsymbol{I} + O(\delta r^2)$. Because the upper triangle part of $\boldsymbol{B}(\theta)$ is the negative of the transpose of the lower triangle part (the diagonal elements are zeros), in fact we only need to learn its lower triangle part. The geometric interpretation is that, if the agent moves along the direction $\theta$, the vector $\boldsymbol{v}(\boldsymbol{x})$ is rotated by the matrix $\boldsymbol{B}(\theta)$, while the $\ell_2$ norm $\|\boldsymbol{v}(\boldsymbol{x})\|^2$ remains stable (figure 1c). We may interpret $\|\boldsymbol{v}(\boldsymbol{x})\|^2 = \sum_{i=1}^d v_i(\boldsymbol{x})^2$ as the total energy of grid cells, which is stable across different locations. From embedding point of view, the local polar system in figure 1a is embedded into a $d$-dimensional sphere in neural response space.

When the agent makes an infinitesimal change of direction from $\theta$ to $\theta + \delta\theta$, $\boldsymbol{B}(\theta)$ is changed to $\boldsymbol{B}(\theta + \delta\theta)$. We assume

$$\boldsymbol{B}(\theta + \delta\theta) = (\boldsymbol{I} + \boldsymbol{C}\delta\theta)\boldsymbol{B}(\theta) + o(\delta\theta), \tag{2}$$

where $\boldsymbol{C}$ is a $d$-dimensional matrix, which is also to be learned. We again assumes $\boldsymbol{C} = -\boldsymbol{C}^\top$, so that $\boldsymbol{I} + \boldsymbol{C}\delta\theta$ is a rotation matrix. The geometric interpretation is that if the agent changes direction, $\boldsymbol{B}(\theta)$ is rotated by $\boldsymbol{C}$.

Equations (1) and (2) together define our proposed model for path integration, which couples two rotation systems.

## 2.2 A MIRROR OF EGOCENTRIC MOTION: PRESERVING LOCAL GEOMETRIC RELATIONS

As a representational model, equations (1) and (2) form a mirror in the $d$-dimensional "mental" (or neural) space for the egocentric motion in the 2D physical space. Importantly, the embedding preserves the local geometric relations of the local polar system.

Let $\delta_\theta v(x) = v(x + \delta x) - v(x)$ be the displacement of $v(x)$ when the agent moves from $x$ by $\delta r$ along direction $\theta$. It follows from equation (1) that $\delta_\theta v(x) = (B(\theta)\delta r + o(\delta r))v(x)$. Ignoring high order terms, we obtain $\delta_{\theta + \delta \theta} v(x) = (I + C\delta\theta)\delta_\theta v(x)$. That is, with $\delta r$ fixed, the local change of $v(x)$ along different $\theta$ are rotated versions of each other, mirroring the local polar system at $x$. See figure 1 for an illustration. As for the angle between $v(x)$ and $v(x + \delta x)$, i.e., how much the vector $v$ rotates in the neural space as the agent moves in 2D physical space by $\delta r$, we have

**Proposition 1** *In the above notation, let $\delta\alpha$ be the angle between $v(x)$ and $v(x + \delta x)$, we have $\delta\alpha = \beta\delta r + O(\delta r^2)$, where $\delta r = \|\delta x\|$, and $\beta = \|B(\theta)v(x)\|^2/\|v(x)\|^2$ is independent of $\theta$.*

See Supplementary A.1 for a proof. That means, the angle $\delta\alpha$ in the $d$-dimensional neural space is proportional to the Euclidean distance $\delta r$ in the 2D space, and more importantly the angle $\delta\alpha$ is independent of direction $\theta$, i.e., $\beta$ is isotropic.

## 2.3 HEXAGON GRID PATTERNS

For the learned model, $\beta$ can be much bigger than 1, so that the vector $v(x)$ will rotate back to itself in a short distance, causing the periodic patterns of $v(x)$. Moreover, $\beta$ does not depend on the direction of self-motion, and this isotropic property appears to underly the emergent hexagonal periodic patterns, as suggested by the following result.

The hexagon grid patterns can be created by linearly mixing three Fourier plane waves whose directions are $2\pi/3$ apart. In the following, we state a theoretical result adapted from Gao et al. (2018) that connects such linearly mixed Fourier waves to the geometric property in Proposition 1 in the previous subsection.

**Proposition 2** *Let $e(x) = (\exp(i\langle a_j, x \rangle), j = 1, 2, 3)^\top$, where $(a_j, j = 1, 2, 3)$ are three 2D vectors of equal norm, and the angle between every pair of them is $2\pi/3$. Let $v(x) = Ue(x)$, where $U$ is an arbitrary unitary matrix, i.e., $U^*U = I$. Let $\delta\alpha$ be the angle between $v(x)$ and $v(x + \delta x)$, we have $\delta\alpha = \beta\delta r + O(\delta r^2)$, where $\delta r = \|\delta x\|$, and $\beta \propto \|a_j\|$ is independent of the direction of $\delta x$.*

See Supplementary A.1 for a proof, which relies on the fact that $(a_j, j = 1, 2, 3)$ forms a tight frame in 2D. Proposition 2 says that the geometric property that emerges from our model as elucidated by Proposition 1 is satisfied by the orthogonal mixing of three Fourier plane waves that creates hexagonal grid patterns. We are currently pursuing a more general analysis of our model, i.e., equations (1) and (2).

## 2.4 JUSTIFICATION AS MINIMALLY SIMPLE RECURRENT MODEL

Now we justify equation 1 as a minimally simple recurrent model. To start, the general form of the model is $v(x + \delta x) = F(v(x), \delta r, \theta)$. For infinitesimal $\delta r$, a first-order Taylor expansion gives

$$v(x + \delta x) = v(x) + f(v(x), \theta)\delta r + o(\delta r), \tag{3}$$

where the function $F(v, \delta, \theta)$ satisfies $F(v, 0, \theta) = v$, i.e., the vector representation stays the same if there is no self-displacement, and $f(v, \theta) = \frac{\partial}{\partial\delta}F(v, \delta, \theta)\mid_{\delta=0}$, i.e., the first derivative at $\delta = 0$.

The function $f(v, \theta)$ transforms $v$ to another vector of the same dimension and the transformation depends on $\theta$. A minimally simple model is a linear transformation that depends on $\theta$, i.e., we can assume $v(x + \delta x) = v(x) + B(\theta)v(x)\delta r + o(\delta r)$, which leads to equation 1, where the linear transformation is $B(\theta)$. Equation 2 can be similarly justified.

In this paper, we assume a linear recurrent model for its simplicity and its explicit geometric meaning as rotation. It is important to emphasize that our arguments are not mutually exclusive with the work based on non-linear recurrent neural network model (Burak & Fiete, 2009; Couey et al., 2013).

In fact, our linear rotation model may serve as a prototype approximation which may help better understand these nonlinear models - a direction we did not pursue here.

## 2.5 MATRIX LIE ALGEBRAS AND GROUPS: FROM INFINITESIMAL TO FINITE

For a finite, non-infinitesimal, self-displacement $\Delta r$, we can divide $\Delta r$ into $N$ steps, so that $\delta r = \Delta r/N \to 0$ as $N \to \infty$, and

$$\boldsymbol{v}(\boldsymbol{x} + \Delta \boldsymbol{x}) = (\boldsymbol{I} + \boldsymbol{B}(\theta)(\Delta r/N) + o(1/N))^N \boldsymbol{v}(\boldsymbol{x}) \to \exp(\boldsymbol{B}(\theta)\Delta r)\boldsymbol{v}(\boldsymbol{x}). \tag{4}$$

The above math underlies the relationship between matrix Lie algebra and matrix Lie group (Taylor, 2002). For a fixed $\theta$, the set of $\boldsymbol{M}_\theta(\Delta r) = \exp(\boldsymbol{B}(\theta\Delta r))$ for $\Delta r \in \mathbb{R}$ forms a matrix Lie group, which is both a group and a manifold. The tangent space of $\boldsymbol{M}_\theta(\Delta r)$ at identity $\boldsymbol{I}$ is called matrix Lie algebra. $\boldsymbol{B}(\theta)$ is the basis of this tangent space, and is often referred to as the generator matrix.

Similarly for a finite change of direction $\Delta \theta$, we obtain

$$\boldsymbol{B}(\theta + \Delta \theta) = \exp(\boldsymbol{C}\Delta\theta))\boldsymbol{B}(\theta). \tag{5}$$

The set of $\boldsymbol{R}(\theta) = \exp(\boldsymbol{C}\Delta\theta)$ for all $\theta \in [0, 2\pi]$ (with mod $2\pi$ addition arithmetics) forms another matrix Lie group, with $\boldsymbol{C}$ being the generator matrix of its matrix Lie algebra.

**Approximation to exponential map**. For a finite but small $\Delta r$, $\exp(\boldsymbol{B}(\theta)\Delta r)$ can be approximated by a second-order Taylor expansion

$$\exp(\boldsymbol{B}(\theta)\Delta r) = \boldsymbol{I} + \boldsymbol{B}(\theta)\Delta r + \boldsymbol{B}(\theta)^2 \Delta r^2/2 + o(\Delta r^2). \tag{6}$$

Similarly, $\exp(\boldsymbol{C}\Delta\theta)$ can be approximated by $\exp(\boldsymbol{C}\Delta\theta) = \boldsymbol{I} + \boldsymbol{C}\Delta\theta + \boldsymbol{C}^2\Delta\theta^2/2 + o(\Delta\theta^2)$.

**Path integration**. Now we can cast path integration in the language of Lie group. Specifically, the input includes the initial position $\boldsymbol{x}^{(0)}$, and the self-motions $(\theta^{(t)}, \Delta r^{(t)})$ for $t = 1, ..., T$. Initializing $\boldsymbol{v}^{(0)} = \boldsymbol{v}(\boldsymbol{x}^{(0)})$, the vector is updated recurrently according to

$$\boldsymbol{v}^{(t)} = \exp(\boldsymbol{B}(\theta^{(t)})\Delta r^{(t)})\boldsymbol{v}^{(t-1)}. \tag{7}$$

That is, the vector $\boldsymbol{v}^{(t)}$ is rotated by $\boldsymbol{B}(\theta^{(t)})$ according to $\Delta r^{(t)}$ geometrically.

**Modules**. Experimentally, it is well established that grid cells are organized in discrete modules (Barry et al., 2007; Stensola et al., 2012) or blocks. We thus partition the vector $\boldsymbol{v}(\boldsymbol{x})$ into $K$ blocks, $\boldsymbol{v}(\boldsymbol{x}) = (\boldsymbol{v}_k(\boldsymbol{x}), k = 1, ..., K)$. Correspondingly the generator matrices $\boldsymbol{B}(\theta) = \mathrm{diag}(\boldsymbol{B}_k(\theta), k = 1, ..., K)$ and $\boldsymbol{C} = \mathrm{diag}(\boldsymbol{C}_k, k = 1, ..., K)$ are block diagonal. This greatly reduces the number of parameters to be learned. Note that each sub-vector $\boldsymbol{v}_k(\boldsymbol{x})$ is rotated by a sub-matrix $\boldsymbol{B}_k(\theta)$, which is in turn rotated by $\boldsymbol{C}_k$.

**Metric**. By the same argument as in Proposition 1, for a module $k$, let $\boldsymbol{v}_k$ be the sub-vector. Let $\delta\alpha_k$ be the angle between $\boldsymbol{v}_k(\boldsymbol{x})$ and $\boldsymbol{v}_k(\boldsymbol{x} + \delta\boldsymbol{x})$, then the angle $\delta\alpha_k = \beta_k\delta r$, where $\delta r = \|\delta\boldsymbol{x}\|$, and $\beta_k$ is independent of $\theta$. That is, if the agent moves by $\delta r$, the vector $\boldsymbol{v}_k(\boldsymbol{x})$ rotates by an angle $\beta_k\delta r$. $\beta_k$ determines the metric or scale of the response maps of the $k$-th block of grid cells, i.e., it tells us how fast the sub-vector $\boldsymbol{v}_k$ rotates as the agent moves.

## 3 INTEGRATION WITH BASIS EXPANSION MODEL

For each $\boldsymbol{v}(\boldsymbol{x})$, we need to uniquely decode $\boldsymbol{x}$. We thus need to integrate the path integration model with the basis expansion model that connects grid cells to place cells. Each place cell fires when the agent is at a specific position. Let $A_{\boldsymbol{x}'}(\boldsymbol{x})$ be the response map for the place cell associated with position $\boldsymbol{x}'$. It measures the adjacency between $\boldsymbol{x}$ and $\boldsymbol{x}'$. A commonly used form of $A_{\boldsymbol{x}'}(\boldsymbol{x})$ when the agent navigates in the open field is the Gaussian adjacency kernel $A_{\boldsymbol{x}'}(\boldsymbol{x}) = \exp(-\|\boldsymbol{x} - \boldsymbol{x}'\|^2/(2\sigma^2))$.

### 3.1 BASIS EXPANSION

A popular model that connects place cells and grid cells is the following basis expansion model (or PCA-based model) (Dordek et al., 2016):

$$A_{\boldsymbol{x}'}(\boldsymbol{x}) = \sum_{i=1}^{d} u_{i,\boldsymbol{x}'} v_i(\boldsymbol{x}) = \langle \boldsymbol{v}(\boldsymbol{x}), \boldsymbol{u}(\boldsymbol{x}') \rangle, \tag{8}$$

where $\boldsymbol{v}(\boldsymbol{x}) = (v_i(\boldsymbol{x}), i = 1, ..., d)^\top$, and $\boldsymbol{u}(\boldsymbol{x}') = (u_{i,\boldsymbol{x}'}, i = 1, ..., d)^\top$. Here $(v_i(\boldsymbol{x}), i = 1, ..., d)$ forms a set of $d$ basis functions for expanding $A_{\boldsymbol{x}'}(\boldsymbol{x})$ for all places $\boldsymbol{x}'$, while $\boldsymbol{u}(\boldsymbol{x}')$ is the read-out weight vector for place cell $\boldsymbol{x}'$, and needs to be learned.

Experimental results have shown that the connections from grid cells to place cells are excitatory (Zhang et al., 2013; Rowland et al., 2018). We thus assume that $u_{i,\boldsymbol{x}'} \geq 0$ for all $i$ and $\boldsymbol{x}'$. We can also make $\boldsymbol{v}(\boldsymbol{x})$ to be non-negative by adding a bias term. Please see Supplementary A.4 for details.

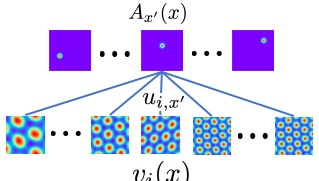

Figure 2: Illustration of basis expansion model $A_{\boldsymbol{x}'}(\boldsymbol{x}) = \sum_{i=1}^{d} u_{i,\boldsymbol{x}'} v_i(\boldsymbol{x})$, where $v_i(\boldsymbol{x})$ is the response map of $i$-th grid cell, shown at the bottom, which shows 5 different $i$. $A_{\boldsymbol{x}'}(\boldsymbol{x})$ is the response map of place cell associated with $\boldsymbol{x}'$, shown at the top, which shows 3 different $\boldsymbol{x}'$. $u_{i,\boldsymbol{x}'}$ is the connection weight.

## 3.2 DECODING, RE-ENCODING, AND ERROR CORRECTION

For a neural response vector $\boldsymbol{v}$, such as $\boldsymbol{v}^{(t)}$ in equation (7), the response of the place cell centered at location $\boldsymbol{x}'$ is $\langle \boldsymbol{v}, \boldsymbol{u}(\boldsymbol{x}') \rangle$. We can decode the position $\hat{\boldsymbol{x}}$ by examining which place cell has the maximal response, i.e.,

$$\hat{\boldsymbol{x}} = \arg\max_{\boldsymbol{x}'} \langle \boldsymbol{v}, \boldsymbol{u}(\boldsymbol{x}') \rangle. \tag{9}$$

After decoding $\hat{\boldsymbol{x}}$, we can re-encode $\boldsymbol{v} \leftarrow \boldsymbol{v}(\hat{\boldsymbol{x}})$, which amounts to projecting $\boldsymbol{v}$ onto the 2D manifold formed by $\boldsymbol{v}(\boldsymbol{x})$ for all $\boldsymbol{x}$. The set of $\boldsymbol{v}(\boldsymbol{x})$ forms a codebook, and the projection via re-encoding enables error correction by removing the possible errors or noises in $\boldsymbol{v}$ (see Supplementary A.2).

## 3.3 UNITARY REPRESENTATION AND HARMONIC ANALYSIS

Underlying the integration of our proposed path integration model (equation 1) and (equation 2) and the basis expansion model (i.e., equation 8) is the group representation theory.

Our path integration model leads to unitary group representation. Let $\boldsymbol{M}_\theta(\Delta r) = \exp(\boldsymbol{B}(\theta)\Delta r)$ with finite (non-infinitesimal) $\Delta r$, let $\boldsymbol{x} = (\Delta r \cos\theta, \Delta r \sin\theta)$, and $\boldsymbol{M}(\boldsymbol{x}) = \boldsymbol{M}_\theta(\Delta r)$. For each given $\boldsymbol{x}$, $\boldsymbol{M}(\boldsymbol{x})$ is an orthogonal matrix. Collectively, $\boldsymbol{M}(\boldsymbol{x})$ forms a unitary representation of $\boldsymbol{x} \in \mathbb{R}^2$, i.e., 2D Euclidean group, where the additive group action in $\mathbb{R}^2$ is represented by matrix multiplication. For each element of the matrix, $M_{ij}(\boldsymbol{x})$ is a function of $\boldsymbol{x}$. According to the fundamental theorems of Schur (Zee, 2016) and Peter-Weyl (Taylor, 2002), if $\boldsymbol{M}$ is an irreducible representation of a finite group or compact Lie group, then $\{M_{ij}(\boldsymbol{x})\}$ form a set of orthogonal basis functions of $\boldsymbol{x}$. This leads to a deep generalization of harmonic analysis or Fourier analysis. Let $\boldsymbol{v}(\boldsymbol{x}) = \boldsymbol{M}(\boldsymbol{x})\boldsymbol{v}(0)$ (where we choose the origin 0 as the reference point). The elements of $\boldsymbol{v}(\boldsymbol{x})$, i.e., $(v_i(\boldsymbol{x}), i = 1, ..., d)$, are also basis functions of $\boldsymbol{x}$. These basis functions serve to expand $(A_{\boldsymbol{x}'}(\boldsymbol{x}), \forall \boldsymbol{x}')$ that parametrizes the place cells, and these basis functions are generated by the matrix Lie algebras of our path integration model. Thus group representation provides a unifying theoretical framework for the two hypothesized roles of grid cells, namely path integration and basis expansion.

In our work, we do not assume each block matrix $\boldsymbol{M}_k(\boldsymbol{x})$ to be irreducible. Thus each learned $v_i(\boldsymbol{x})$ within a block is a linear mixing of orthogonal basis functions in an irreducible representation, and different $v_i(\boldsymbol{x})$ within the same block are not necessarily orthogonal. However, different $v_i(\boldsymbol{x})$ in different blocks are close to orthogonal in our experiments (see Supplementary A.3 for details).

## 4 LEARNING

The unknown parameters are (1) $(\boldsymbol{v}(\boldsymbol{x}), \forall \boldsymbol{x})$. (2) $(\boldsymbol{u}(\boldsymbol{x}'), \forall \boldsymbol{x}')$. (3) $(\boldsymbol{B}(\theta), \forall \theta)$. (4) $\boldsymbol{C}$. To learn these parameters, we define a loss function: $L = L_0 + \lambda_1 L_1 + \lambda_2 L_2$, where

$$L_0 = \mathbb{E}_{\boldsymbol{x},\boldsymbol{x}'}[A_{\boldsymbol{x}'}(\boldsymbol{x}) - \langle \boldsymbol{v}(\boldsymbol{x}), \boldsymbol{u}(\boldsymbol{x}') \rangle]^2, \tag{10}$$

$$L_1 = \mathbb{E}_{\boldsymbol{x},\Delta\boldsymbol{x}}\|\boldsymbol{v}(\boldsymbol{x} + \Delta\boldsymbol{x}) - \exp(\boldsymbol{B}(\theta)\Delta r)\boldsymbol{v}(\boldsymbol{x})\|^2, \tag{11}$$

$$L_2 = \mathbb{E}_{\theta,\Delta\theta}\|\boldsymbol{B}(\theta + \Delta\theta) - \exp(\boldsymbol{C}\Delta\theta)\boldsymbol{B}(\theta)\|^2, \tag{12}$$

where $\|\cdot\|^2$ denotes the sum of squares of the elements of the vector or matrix. $\lambda_1$ and $\lambda_2$ are chosen so that the three loss terms are of similar magnitudes. In $L_0$, $A_{\boldsymbol{x}'}(\boldsymbol{x})$ are given as Gaussian adjacency kernels, and we aim to learn the basis functions $\boldsymbol{v}(\boldsymbol{x})$. $L_1$ and $L_2$ serve to constrain

the basis functions $\boldsymbol{v}(\boldsymbol{x})$ so that path integration can be performed based on our proposed model (equation 1) and (equation 2). In $L_1$, $\Delta \boldsymbol{x} = (\Delta r \cos \theta, \Delta r \sin \theta)$.

As the generator matrices for two coupled rotation systems, $\boldsymbol{B}(\theta)$ and $\boldsymbol{C}$ are both assumed to be skew-symmetric, so that only the lower triangle parts of the matrices need to be learned. We further assume $\boldsymbol{B}(\theta)$ and $\boldsymbol{C}$ to be block-diagonal with each block corresponding to a module, consistent with the experimental observations (Stensola et al., 2012). For regularization, we add a penalty on $|\boldsymbol{u}(\boldsymbol{x}')|^2$, and further assume $\boldsymbol{u}(\boldsymbol{x}') \geq 0$ so that the connections from grid cells to place cells are excitatory (Zhang et al., 2013; Rowland et al., 2018).

We minimize the loss function by stochastic gradient descent, specifically, *Adam* optimizer (Kingma & Ba, 2014), where the expectations are approximated by Monte Carlo samples of $(\boldsymbol{x}, \boldsymbol{x}')$ and $\boldsymbol{x}, \Delta \boldsymbol{x}$. See Supplementary B.1 for details of generating Monte Carlo samples for learning.

Unlike previous work on learning basis expansion model (or PCA-based model (Dordek et al., 2016)), we do not constrain the basis functions $\boldsymbol{v}(\boldsymbol{x}) = (v_i(\boldsymbol{x}), i = 1, ..., d)$ to be orthogonal to each other. Instead, we constrain them by our path integration model (1) and (2) via the loss terms $L_1$ and $L_2$. In fact, the learned $v_i(\boldsymbol{x})$ within the same block are not orthogonal, although $v_i(\boldsymbol{x})$ from different blocks tend to be orthogonal (see Supplementary A.3). $L_2$ constrains $\boldsymbol{B}(\theta)$ to be rotated versions of each other, and it is important for the emergence of hexagon grid patterns according to our experiments. It is also important for reducing model complexity by constraining $\boldsymbol{B}(\theta)$ for different $\theta$.

Because $A_{\boldsymbol{x}'}(\boldsymbol{x})$ contains a whole range of frequencies in the Fourier domain, the learned response maps of the grid cells span a range of scales too. It is also worth noting that, consistent with the experiential observations, we assume individual place field $A_{\boldsymbol{x}'}(\boldsymbol{x})$ to exhibit a Gaussian shape, rather than a Mexican-hat pattern (with balanced excitatory center and inhibitory surround) as assumed in previous basis expansion models (Dordek et al., 2016; Sorscher et al., 2019) of grid cells.

## 5 EXPERIMENTS

In the numerical experiments, we use a square environment with size 2m × 2m, which is discretized into a 80 × 80 lattice. For direction, we discretize the circle $[0, 2\pi]$ into 144 directions. With the above discretizations, we use nearest neighbor linear interpolations for values in between. We use the second-order Taylor expansion (6) to approximate the exponential maps $\exp(\boldsymbol{B}(\theta)\Delta r)$ and $\exp(\boldsymbol{C}\Delta\theta)$. The local motion $\Delta r$ and $\Delta \theta$ are constrained to local ranges, i.e., $\Delta r$ is smaller than 3 grids on the lattice and $\Delta \theta$ is smaller than 12.5 degrees. For the adjacency kernel $A_{\boldsymbol{x}'}(\boldsymbol{x})$ in $L_0$ in equation 10, we use a Gaussian kernel with $\sigma = 0.07$. $\boldsymbol{v}(\boldsymbol{x})$ is of $d = 192$ dimensions, which is partitioned into $K = 6$ modules, each of which has 32 cells. Our results are robust to the specific choice of the number of modules or the number of cells (see Supplementary B.2).

### 5.1 HEXAGON GRID PATTERNS

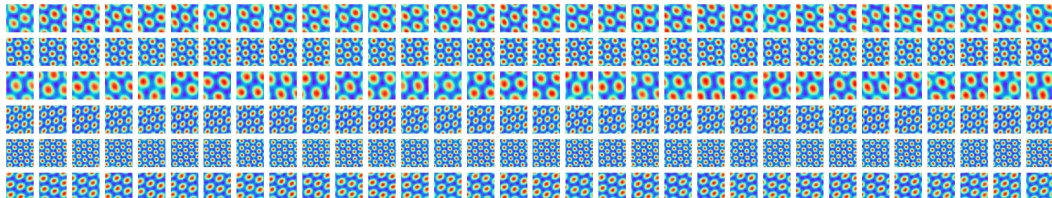

Figure 3: Grid firing patterns emerge in the learned network. Every response map shows the firing pattern of one neuron (i.e, one element of $\boldsymbol{v}$) in the 2D environment. Every row shows the firing patterns of the neurons within the same block or module. (Zoom in for high quality.)

Figure 3 shows the learned firing pattern $\boldsymbol{v}(\boldsymbol{x}) = (v_i(\boldsymbol{x}), i = 1, ..., d)$ over the 80 × 80 lattice of $\boldsymbol{x}$. Every row shows the learned units belonging to the same block or module. Regular hexagon grid patterns emerge for both $\boldsymbol{v}(\boldsymbol{x})$ and $\boldsymbol{u}(\boldsymbol{x})$. Within each block or module, the scales and orientations are roughly the same, but with different phases or spatial shifts. Notably, the emergence of hexagon patterns does not rely on specific block size nor number of blocks. See Supplementary B.2 for the

patterns of learned $\boldsymbol{v}(\boldsymbol{x})$ with different block sizes, learned $\boldsymbol{B}(\theta)$ and learned $\boldsymbol{u}(\boldsymbol{x})$. For the learned $\boldsymbol{B}(\theta)$, each element shows regular sine/cosine tuning over $\theta$.

We further investigate the characteristics of the learned firing rate patterns (i.e., $\boldsymbol{v}(\boldsymbol{x})$) using measures adopted from the grid cell literature. Specifically, the hexagonal regularity, scale and orientation of grid-like patterns are quantified using the gridness score, grid scale and grid orientation (Langston et al., 2010; Sargolini et al., 2006), which are determined by taking a circular sample of the auto-correlogram of the response map. All learned patterns exhibit significant hexagonal periodicity in terms of gridness scores (mean $1.08$, range $0.60$ to $1.57$). Specifically, a unit is considered to be grid-like if the gridness score exceeds the $95^{th}$ percentile of null distribution obtained by applying spatial field shuffles to the response map, following the standard procedure in (Hafting et al., 2005; Barry & Burgess, 2017). On average, the $95^{th}$ percentile is $0.35$ for all the units. Figure 4a shows six examples of the autocorrelograms of the response maps and the corresponding gridness scores, each of which is from a different module. The grid scales of learned patterns (mean $0.39$, range $0.24$ to $0.61$), as shown in Figure 4b, follows a multi-modal distribution. The ratio between neighboring modes are roughly $1.44$ and $1.51$, which closely matches the theoretical predictions (Wei et al., 2015; Stemmler et al., 2015) and also the empirical results from rodent grid cells (Stensola et al., 2012). The grid orientations of learned patterns, as shown in Figure 4c, are also multi-modal distributed, consistent to the observations on rat grid cells (Stensola et al., 2012). See Supplementary B.3 for the detailed spatial profile of every unit of $\boldsymbol{v}(\boldsymbol{x})$. Collectively, these results reveal striking, quantitative correspondence between the properties of our model neurons and those of the grid cells in the brain.

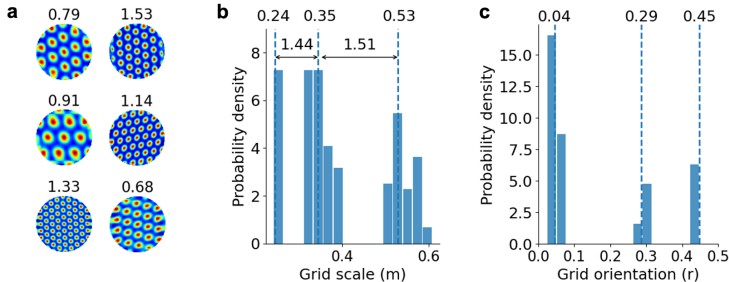

Figure 4: Model grid cells exhibit modular structure that is consistent with experimental data.(a) Examples of autocorrelograms of the response maps and the corresponding gridness scores, each of which is from a different module. (b) Multi-modal distribution of grid scales. The scale ratios closely match the real data (Stensola et al., 2012). (c) Multi-modal distribution of grid orientations.

## 5.2 PATH INTEGRATION AND ERROR CORRECTION

We then examine the ability of the learned system on performing path integration, by recurrently updating $\boldsymbol{v}^{(t)}$ as shown in equation 7 and decoding $\boldsymbol{v}^{(t)}$ to $\boldsymbol{x}^{(t)}$ for $t = 1, ..., T$ using equation 9. Re-encoding $\boldsymbol{v}^{(t)} \leftarrow \boldsymbol{v}(\boldsymbol{x}^{(t)})$ after decoding is adopted. Figure 5a shows an example trajectory of accurate path integration for $T = 80$. As shown in figure 5b, with re-encoding, the path integration error remains close to zero over a duration of $500$ time steps ($< 0.01$ cm, averaged over $1,000$ episodes), although the model is trained by the single-time-step loss in equation 11. Without re-encoding, the error goes slight higher but still remains reasonable (ranging from $0.0$ to $5.4$ cm, mean $3.8$ cm). The performance of path integration would be improved as the block size becomes larger, i.e., more units or cells in each module (figure 5c). When block size is larger than $20$, path integration is almost exact for the time steps tested.

We further assess the ability of error correction of the learned system. Specifically, along the way of path integration, at every time step $t$, two types of errors are introduced to $\boldsymbol{v}^{(t)}$: (1) Gaussian noise or (2) dropout masks, i.e., certain percentage of units are randomly set to zero. Figure 5d summarizes the path integration performance with different levels of introduced errors for $T = 100$. For Gaussian noise, we use the average magnitude of units in $\boldsymbol{v}(\boldsymbol{x})$ as the reference standard deviation ($s$), i.e., $s = \sqrt{|\boldsymbol{v}(\boldsymbol{x})|^2/d}$. The results show that re-encoding is crucial for error correction. Notably, with re-encoding, the path integration works reasonably well even if Gaussian noise with magnitude of $s$ is added or $50\%$ units are dropped out at each step, indicating that the learned system is quite robust to different sources of errors.

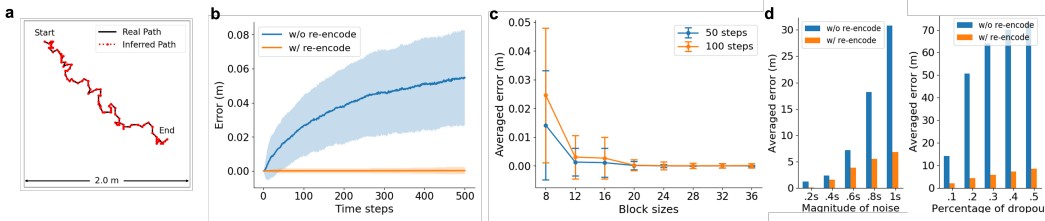

Figure 5: The learned model can perform path integration. (a) Black: example trajectory. The decoded self-positions (red) accurately matches the real path. (b) Path integration error over number of time steps. (c) Path integration error over different block sizes, for 50 and 100 time steps. For (b) and (c), averaged error and $\pm$ 1 standard deviation band over 1,000 episodes are shown. (d) Path integration error with introduced errors. *Left*: Gaussian noise. *Right*: dropout mask.

### 5.3 ABLATION STUDY

We conduct systematic ablation study to assess the impact of individual components of the model on learning hexagonal grid patterns (measured by gridness score) and the ability to perform path integration ($T = 100$ time steps). Table 1 summarizes the results where the model is trained with certain component removed. Specifically, for the emergence of regular hexagonal patterns, all components appear to be important, especially $L_2$ which is crucial. In comparison, $\boldsymbol{u}(\boldsymbol{x}) \geq 0$ is not entirely critical. Another observation is that $L_0$, $L_1$ and penalty on $|\boldsymbol{u}(\boldsymbol{x})|^2$ are crucial for accurate path integration. See Supplementary B.4 for details.

Table 1: Ablation study. A certain component of the model is removed and the learned model is evaluated in terms of gridness score and path integration error over $T = 100$ time steps. Skew-symmetry is for $\boldsymbol{B}(\theta)$ and $\boldsymbol{C}$.

| $L_0$ | $L_1$ | $L_2$ | Regularize $|\boldsymbol{u}(\boldsymbol{x})|^2$ | Skew-symmetry | $\boldsymbol{u}(\boldsymbol{x}) \geq 0$ | Gridness | path integration error (cm) |
|---|---|---|---|---|---|---|---|
| ✗ | ✓ | ✓ | ✓ | ✓ | ✓ | $-0.10 \pm 0.15$ | $44.2 \pm 15.4$ |
| ✓ | ✗ | ✓ | ✓ | ✓ | ✓ | $0.23 \pm 0.22$ | $30.6 \pm 11.8$ |
| ✓ | ✓ | ✗ | ✓ | ✓ | ✓ | $0.32 \pm 0.31$ | $0.00 \pm 0.00$ |
| ✓ | ✓ | ✓ | ✗ | ✓ | ✓ | $-0.07 \pm 0.21$ | $26.3 \pm 10.2$ |
| ✓ | ✓ | ✓ | ✓ | ✗ | ✓ | $0.70 \pm 0.41$ | $0.00 \pm 0.00$ |
| ✓ | ✓ | ✓ | ✓ | ✓ | ✗ | $0.89 \pm 0.27$ | $0.00 \pm 0.00$ |
| ✓ | ✓ | ✓ | ✓ | ✓ | ✓ | $1.08 \pm 0.31$ | $0.00 \pm 0.00$ |

## 6 DISCUSSION AND CONCLUSION

### 6.1 RELATED WORK ON MODELS OF GRID CELLS

Our work is related to several lines of previous research. First, RNN models have been used to model grid cells and path integration. The traditional approach uses simulation-based models with hand-crafted connectivity (Zhang, 1996; Burak & Fiete, 2009; Couey et al., 2013; Pastoll et al., 2013; Agmon & Burak, 2020). More recently, two pioneering papers (Cueva & Wei, 2018; Banino et al., 2018) developed an optimization-based RNN approach to learn the path integration model and discovered that grid-like response pattern could emerge in the optimized network. These results are further substantiated in following-up research (Sorscher et al., 2019; Cueva et al., 2020). Compared to these studies, our path integration model is more explicit, coupling two rotation systems in neural space to mirror the egocentric motion in physical space. In doing so, our results reveals new insights regarding why lattice and in particular hexagonal response patterns may emerge in neural networks trained to perform path integration.

Second, as discussed in Sec.3, our model is naturally connected to the basis expansion models of grid cells. A key ingredient of this line of work (Dordek et al., 2016; Sorscher et al., 2019; Stachenfeld et al., 2017) is that, under certain conditions, the principal components of the place cell activities exhibit grid patterns. Importantly, our work differs from these models in that, unlike PCA, we make no

assumption about the orthogonality between the basis functions, and the basis expansion formulation is obtained via group representation from our path integration model. Group representation unifies path integration and basis expansion, which are two roles hypothesized for grid cells. Furthermore, in previous basis expansion models (Dordek et al., 2016; Sorscher et al., 2019), place fields with Mexican-hat patterns (with balanced excitatory center and inhibitory surround) were assumed in order to obtain hexagonal grid firing patterns. However, experimentally measured place fields were instead well characterized by Gaussian functions. Crucially, in our model, hexagonal grids emerge after learning with Gaussian place fields, and there is no need to assume any additional surround mechanisms.

In another related paper, Gao et al. (2018) proposed vector representation of 2D position and matrix representation of 2D displacement. Our work goes beyond that work in revealing two coupled Lie groups and Lie algebra structure, as well as further integrating the path integration model with the basis expansion model.

## 6.2 CONCLUSION

This paper entertains a representation model of grid cells' path integration, which mirrors the egocentric self-motion by coupling two rotation systems. The proposed model can be justified as a minimally simple recurrent model, and has explicit geometric and algebraic structures. As we have shown, this simple model framework leads to a system that captures many of the experimentally observed properties of the grid cell system in the brain.

Our path integration model is linear in the vector $v(x)$, but it is non-linear in the displacement $\Delta x$. Since the basis expansion model is linear in $v(x)$, it may be preferable to have the path integration model linear in $v(x)$ too. The rotation of $v(x)$ is capable of path integration, and the way $v(x)$ rotates can explain the hexagon periodic patterns and the metric of each module. The connection between the two models is based on the group representation theory.

As to modularity, in the context of our path integration model, it means that each sub-vector $v_k(x)$ is rotated by a separate generator sub-matrix, or driven by a separate recurrent network, so that the dynamics of the sub-vectors are disentangled. This appears to be biologically plausible. We believe modularity is part of the design of the network.

In terms of representation learning, it is common to represent the physical state by a vector, i.e., embedding the physical state into a higher dimensional neural space. However, the problem of representing continuous motion in the physical space, or continuous transformation of the physical state, or continuous relation between the states has not received an in-depth treatment. The continuous motion in the physical space often has a native Lie algebra and Lie group structure. Our work provides an explicit representation of the continuous motion and its algebraic structure by matrix Lie algebra and matrix Lie group that act in the neural space. We believe our method can be applied to modeling head direction system as well as modeling the motor cortex for representing the continuous motions of arm, hand, pose, etc.

Last but not least, the proposed model can also be used for path planning. See Supplementary C for the preliminary results).

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

# A   THEORETICAL ANALYSIS

## A.1   PROOF OF PROPOSITIONS

**Proof of Proposition 1**. For an infinitesimal $\delta \boldsymbol{x} = \delta r(\cos \theta, \sin \theta)$, $\boldsymbol{v}(\boldsymbol{x})$ is changed to $\boldsymbol{v}(\boldsymbol{x} + \delta \boldsymbol{x})$. Let $\delta \alpha$ be the angle between $\boldsymbol{v}(\boldsymbol{x})$ and $\boldsymbol{v}(\boldsymbol{x} + \delta \boldsymbol{x})$. It can be obtained from

$$
\begin{aligned}
\langle \boldsymbol{v}(\boldsymbol{x}), \boldsymbol{v}(\boldsymbol{x} + \delta \boldsymbol{x}) \rangle &= \boldsymbol{v}(\boldsymbol{x})^\top \exp(\boldsymbol{B}(\theta) \delta r) \boldsymbol{v}(\boldsymbol{x}) &(13)\\
&= \boldsymbol{v}(\boldsymbol{x})^\top (\boldsymbol{I} + \boldsymbol{B}(\theta) \delta r + \boldsymbol{B}(\theta)^2 \delta r^2/2 + o(\delta r^2)) \boldsymbol{v}(\boldsymbol{x}) &(14)\\
&= \|\boldsymbol{v}(\boldsymbol{x})\|^2 - \|\boldsymbol{B}(\theta)\boldsymbol{v}(\boldsymbol{x})\|^2 \delta r^2/2 + o(\delta r^2), &(15)
\end{aligned}
$$

where $\boldsymbol{v}(\boldsymbol{x})^\top \boldsymbol{B}(\theta) \boldsymbol{v}(\boldsymbol{x}) = 0$ because $\boldsymbol{B}(\theta) = -\boldsymbol{B}(\theta)^\top$. Let

$$
\beta^2 = \|\boldsymbol{B}(\theta)\boldsymbol{v}(\boldsymbol{x})\|^2 / \|\boldsymbol{v}(\boldsymbol{x})\|^2. \tag{16}
$$

It is independent of $\theta$, because for any $\Delta\theta$, $\boldsymbol{B}(\theta + \Delta\theta)\boldsymbol{v} = \boldsymbol{R}(\Delta\theta)\boldsymbol{B}(\theta)\boldsymbol{v}$, and $\boldsymbol{R}(\Delta\theta)$ is orthogonal, thus $\|\boldsymbol{B}(\theta + \Delta\theta)\boldsymbol{v}\|^2 = \|\boldsymbol{B}(\theta)\boldsymbol{v}\|^2$ for any $\Delta\theta$. Note $\|\boldsymbol{v}(\boldsymbol{x})\|^2 = \|\boldsymbol{v}(\boldsymbol{x} + \delta\boldsymbol{x})\|^2$ because $\boldsymbol{M}_\theta(\delta r)$ is orthogonal,

$$
\begin{aligned}
\cos(\delta\alpha) &= 1 - \delta\alpha^2/2 + o(\delta\alpha^2) &(17)\\
&= \langle \boldsymbol{v}(\boldsymbol{x}), \boldsymbol{v}(\boldsymbol{x} + \delta\boldsymbol{x}) \rangle / |\boldsymbol{v}(\boldsymbol{x})|^2 &(18)\\
&= 1 - (\beta\delta r)^2/2 + o(\delta r^2). &(19)
\end{aligned}
$$

Thus the angle $\delta\alpha = \beta\delta r$. $\square$

**Proof of Proposition 2**. Suppose $\boldsymbol{v}(\boldsymbol{x})$ is defined as in Proposition 2. That is, $\boldsymbol{v}(\boldsymbol{x}) = \boldsymbol{U}\boldsymbol{e}(\boldsymbol{x})$, where $\boldsymbol{U}$ is an arbitrary unitary matrix. $\boldsymbol{e}(\boldsymbol{x}) = (\exp(i\langle \boldsymbol{a}_j, \boldsymbol{x} \rangle), j = 1, 2, 3)^\top$, where $(\boldsymbol{a}_j, j = 1, 2, 3)$ are three 2D vectors of equal norm, and the angle between every pair of them is $2\pi/3$. Then we have $\|\boldsymbol{v}(\boldsymbol{x})\|^2 = 3$, and the angle between $\boldsymbol{v}(\boldsymbol{x})$ and $\boldsymbol{v}(\boldsymbol{x} + \delta\boldsymbol{x})$, denoted as $\delta\alpha$, is

$$
\cos(\delta\alpha) = \mathrm{RE}\left( \frac{\langle \boldsymbol{v}(\boldsymbol{x}), \boldsymbol{v}(\boldsymbol{x} + \delta\boldsymbol{x}) \rangle}{\|\boldsymbol{v}(\boldsymbol{x})\|\|\boldsymbol{v}(\boldsymbol{x} + \delta\boldsymbol{x})\|} \right) \tag{20}
$$

$$
= \frac{1}{3}\mathrm{RE}(\langle \boldsymbol{v}(\boldsymbol{x}), \boldsymbol{v}(\boldsymbol{x} + \delta\boldsymbol{x}) \rangle) \tag{21}
$$

$$
= \frac{1}{3}\mathrm{RE}(\boldsymbol{e}(\boldsymbol{x})^* \boldsymbol{U}^* \boldsymbol{U} \boldsymbol{e}(\boldsymbol{x} + \delta\boldsymbol{x})) \tag{22}
$$

$$
= \frac{1}{3}\mathrm{RE}\left( \sum_{j=1}^3 \exp(i\langle \boldsymbol{a}_j, \delta\boldsymbol{x} \rangle) \right) \tag{23}
$$

$$
= \frac{1}{3} \sum_{j=1}^3 \cos(\langle \boldsymbol{a}_j, \delta\boldsymbol{x} \rangle) \tag{24}
$$

$$
= \frac{1}{3} \sum_{j=1}^3 \left( 1 - \frac{1}{2}\langle \boldsymbol{a}_j, \delta\boldsymbol{x} \rangle^2 \right) + o(\delta r^2) \tag{25}
$$

$$
= \left( 1 - \frac{\beta^2}{2}\delta r^2 \right) + o(\delta r^2) \tag{26}
$$

$$
= \cos(\beta\delta r) + o(\delta r^2). \tag{27}
$$

Thus we have $\delta\alpha = \beta\delta r + O(\delta r^2)$. Here the key is that $(\boldsymbol{a}_j, j = 1, 2, 3)$ forms a tight frame in the 2D space, in that for any 2D vector $\delta\boldsymbol{x}$, $\sum_{j=1}^3 \langle \delta\boldsymbol{x}, \boldsymbol{a}_j \rangle^2 \propto \|\boldsymbol{a}_j\|^2 \|\delta\boldsymbol{x}\|^2$. Thus $\beta \propto \|\boldsymbol{a}_j\|$. $\square$

## A.2   ERROR CORRECTION

Suppose $\boldsymbol{v} = \boldsymbol{v}(\boldsymbol{x}) + \epsilon$ is a noisy version of $\boldsymbol{v}(\boldsymbol{x})$, can we still decode $\boldsymbol{x}$ accurately from $\boldsymbol{v}$? Here we assume $\epsilon \sim \mathcal{N}(0, \tau^2(|\boldsymbol{v}(\boldsymbol{x})|^2/d)\boldsymbol{I})$, where $d$ is the dimensionality of $\boldsymbol{v}$, and $\tau^2$ measures the variance of noise relative to $|\boldsymbol{v}(\boldsymbol{x})|^2/d$, i.e., the average of $(v_i(\boldsymbol{x})^2, i = 1, ..., d)$.

The heat map

$$h(\boldsymbol{x}') = \langle \boldsymbol{v}, \boldsymbol{u}(\boldsymbol{x}') \rangle = \langle \boldsymbol{v}(\boldsymbol{x}), \boldsymbol{u}(\boldsymbol{x}') \rangle + \langle \epsilon, \boldsymbol{u}(\boldsymbol{x}') \rangle = A(\boldsymbol{x}, \boldsymbol{x}') + e(\boldsymbol{x}'), \qquad (28)$$

where $e(\boldsymbol{x}') \sim \mathcal{N}(0, \tau^2 |\boldsymbol{v}(\boldsymbol{x})|^2 |\boldsymbol{u}(\boldsymbol{x}')|^2 / d)$. For $A(\boldsymbol{x}, \boldsymbol{x}') = \exp(-|\boldsymbol{x} - \boldsymbol{x}'|^2 / (2\sigma^2)) = \langle \boldsymbol{v}(\boldsymbol{x}), \boldsymbol{u}(\boldsymbol{x}') \rangle$, if $\sigma^2$ is small, $A(\boldsymbol{x}, \boldsymbol{x}')$ decreases to 0 quickly, i.e., if $|\boldsymbol{x}' - \boldsymbol{x}| > \delta$, then $A(\boldsymbol{x}, \boldsymbol{x}') < \exp(-\delta^2 / (2\sigma^2))$, and the chance for the maximum of $h(\boldsymbol{x}')$ to be achieved at an $\boldsymbol{x}'$ so that $|\boldsymbol{x}' - \boldsymbol{x}| > \delta$ can be very small.

The above analysis also provides a justification for regularizing $|\boldsymbol{u}(\boldsymbol{x}')|^2$ in learning.

For error correction, we want $d$ to be big, and we want $\sigma^2$ to be small. But for path planning, we also need big $\sigma^2$. That is, we need $A(\boldsymbol{x}, \boldsymbol{x}')$ at multiple scales.

### A.3 ORTHOGONALITY RELATIONS

For $(\boldsymbol{x})$ that form a group, a matrix representation $\boldsymbol{M}(\boldsymbol{x})$ is equivalent to another representation $\tilde{\boldsymbol{M}}(\boldsymbol{x})$ if there exists a matrix $\boldsymbol{P}$ such that $\tilde{\boldsymbol{M}}(\boldsymbol{x}) = \boldsymbol{P} \boldsymbol{M}(\boldsymbol{x}) \boldsymbol{P}^{-1}$ for each $\boldsymbol{x}$. A matrix representation is reducible if it is equivalent to a block diagonal matrix representation, i.e., we can find a matrix $\boldsymbol{P}$, such that $\boldsymbol{P} \boldsymbol{M}(\boldsymbol{x}) \boldsymbol{P}^{-1}$ is block diagonal for every $\boldsymbol{x}$. Suppose the group is a finite group or a compact Lie group, and $\boldsymbol{M}$ is a unitary representation. If $\boldsymbol{M}$ is block-diagonal, $\boldsymbol{M} = \text{diag}(\boldsymbol{M}_k, k = 1, ..., K)$, with non-equivalent blocks, and each block $\boldsymbol{M}_k$ cannot be further reduced, then the matrix elements $(\boldsymbol{M}_{kij}(\boldsymbol{x}))$ are orthogonal basis functions of $\boldsymbol{x}$. Such orthogonality relations are proved by Schur (Zee, 2016) for finite group, and by Peter-Weyl for compact Lie group (Taylor, 2002). For our case, theoretically the group of displacements in the 2D domain is $\mathbb{R}^2$, but we learn our model within a finite range, and we further discretize the range into a lattice. Thus the above orthogonal relations are relevant.

In our model, we also assume block diagonal $\boldsymbol{M}$, and we call each block a module. However, we do not assume each module is irreducible, i.e., each module itself may be further diagonalized into a block diagonal matrix of irreducible blocks. Thus the elements within the same module $\boldsymbol{v}_k(\boldsymbol{x})$ may be linear mixing of orthogonal basis functions, and they themselves may not be orthogonal. However, different modules may be linear mixings of different sets of irreducible blocks, and thus different modules can be orthogonal to each other. Figure 6 visualizes the correlations between each pair of the learned $\boldsymbol{v}_i(\boldsymbol{x})$ and $\boldsymbol{v}_j(\boldsymbol{x})$, $i, j = 1, ..., d$. For $\boldsymbol{v}_i(\boldsymbol{x})$ and $\boldsymbol{v}_j(\boldsymbol{x})$ from different modules, the correlations are close to zero; i.e., $\boldsymbol{v}_i(\boldsymbol{x})$ and $\boldsymbol{v}_j(\boldsymbol{x})$ from different blocks are approximately orthogonal to each other. But $\boldsymbol{v}_i(\boldsymbol{x})$ and $\boldsymbol{v}_j(\boldsymbol{x})$ from the same block are not orthogonal to each other.

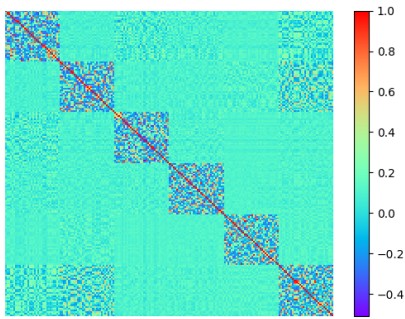

Figure 6: Correlation heatmap for each pair of the learned $v_i(\boldsymbol{x})$ and $v_j(\boldsymbol{x})$. The correlations are computed over $80 \times 80$ lattice of $\boldsymbol{x}$.

### A.4 ADDING BIAS TERM

Since $\|\boldsymbol{v}(\boldsymbol{x})\|$ is fixed under rotation, we can add a bias $\boldsymbol{b}$ to the vector $\boldsymbol{v}(\boldsymbol{x})$ so that the elements become non-negative. Specifically, let $\tilde{\boldsymbol{v}}(\boldsymbol{x}) = \boldsymbol{v}(\boldsymbol{x}) + \boldsymbol{b}$, where $\boldsymbol{b}$ is $d$-dimensional vector with equal

elements, then $\boldsymbol{v}(\boldsymbol{x}) = \tilde{\boldsymbol{v}}(\boldsymbol{x}) - \boldsymbol{b}$. The recurrent model for $\boldsymbol{v}(\boldsymbol{x} + \delta\boldsymbol{x}) = (\boldsymbol{I} + \boldsymbol{B}(\theta)\delta r)\boldsymbol{v}(\boldsymbol{x})$ can then be translated into $\tilde{\boldsymbol{v}}(\boldsymbol{x} + \delta\boldsymbol{x}) = (\boldsymbol{I} + \boldsymbol{B}(\theta)\delta r)\tilde{\boldsymbol{v}}(\boldsymbol{x}) - \boldsymbol{c}\delta r$, where $\boldsymbol{c} = -\boldsymbol{B}(\theta)\boldsymbol{b}$. The basis expansion model $A_{\boldsymbol{x}'}(\boldsymbol{x}) = \langle \boldsymbol{v}(\boldsymbol{x}), \boldsymbol{u}(\boldsymbol{x}') \rangle$ can be translated into $A_{\boldsymbol{x}'}(\boldsymbol{x}) = \langle \tilde{\boldsymbol{v}}(\boldsymbol{x}), \boldsymbol{u}(\boldsymbol{x}') \rangle - b(\boldsymbol{x}')$, where $b(\boldsymbol{x}') = \langle \boldsymbol{b}, \boldsymbol{u}(\boldsymbol{x}') \rangle$. $\tilde{\boldsymbol{v}}(\boldsymbol{x})$ rotates around $\boldsymbol{b}$.

# B    EXPERIMENTS

## B.1    MONTE CARLO SAMPLING

For learning, the expectations in loss terms are approximated by Monte Carlo samples. Here we detail the generation of Monte Carlo samples. For $(\boldsymbol{x}, \boldsymbol{x}')$ used in $L_0 = \mathbb{E}_{\boldsymbol{x}, \boldsymbol{x}'}[A(\boldsymbol{x}, \boldsymbol{x}') - \langle \boldsymbol{v}(\boldsymbol{x}), \boldsymbol{u}(\boldsymbol{x}') \rangle]^2$, $\boldsymbol{x}$ is first sampled uniformly within the entire domain, and then the displacement $d\boldsymbol{x}$ between $\boldsymbol{x}$ and $\boldsymbol{x}'$ is sampled from a normal distribution $\mathcal{N}(0, \sigma^2 \boldsymbol{I}_2)$, where $\sigma = 0.48$. This is to ensure that nearby samples are given more emphasis. We let $\boldsymbol{x}' = \boldsymbol{x} + d\boldsymbol{x}$, and those pairs $(\boldsymbol{x}, \boldsymbol{x}')$ within the range of domain (i.e., 2m × 2m, 80 × 80 lattice) are kept as valid data. For $(\boldsymbol{x}, \Delta\boldsymbol{x})$ used in $L_1 = \mathbb{E}_{\boldsymbol{x}, \Delta\boldsymbol{x}} |\boldsymbol{v}(\boldsymbol{x} + \Delta\boldsymbol{x}) - \exp(\boldsymbol{B}(\theta)\Delta r)\boldsymbol{v}(\boldsymbol{x})|^2$, $\Delta\boldsymbol{x}$ is sampled uniformly within a circular domain with radius equal to 3 grids and $(0, 0)$ as the center. Specifically, $\Delta r^2$, the squared length of $\Delta\boldsymbol{x}$, is sampled uniformly from $[0, 3]$ grids, and $\theta$ is sampled uniformly from $[0, 2\pi]$. We take the square root of the sampled $\Delta r^2$ as $\Delta r$ and let $\Delta\boldsymbol{x} = (\Delta r \cos\theta, \Delta r \sin\theta)$. Then $\boldsymbol{x}$ is uniformly sampled from the region such that both $\boldsymbol{x}$ and $\boldsymbol{x} + \Delta\boldsymbol{x}$ are within the range of domain. For $(\theta, \Delta\theta)$ used in $L_2 = \mathbb{E}_{\theta, \Delta\theta} |\boldsymbol{B}(\theta + \Delta\theta) - \exp(\boldsymbol{C}\Delta\theta)\boldsymbol{B}(\theta)|^2$, we enumerate all the pairs of discretized $\theta$ (i.e., 144 directions discretized for circle $[0, 2\pi]$) and $\Delta\theta$ (i.e., 5 directions within range $[0, 12.5]$ degrees) as samples.

## B.2    LEARNED PATTERNS

Figure 7 shows the learned patterns of $\boldsymbol{u}(\boldsymbol{x})$ with 6 blocks of size 32 cells in each block. Figure 8 shows the learned patterns of $\boldsymbol{v}(\boldsymbol{x})$ and $\boldsymbol{u}(\boldsymbol{x})$ with 6 blocks of size 16. Figure 9 shows the learned patterns of $\boldsymbol{v}(\boldsymbol{x})$ with 16 blocks if size 12. Regular hexagon patterns emerge in all these settings.

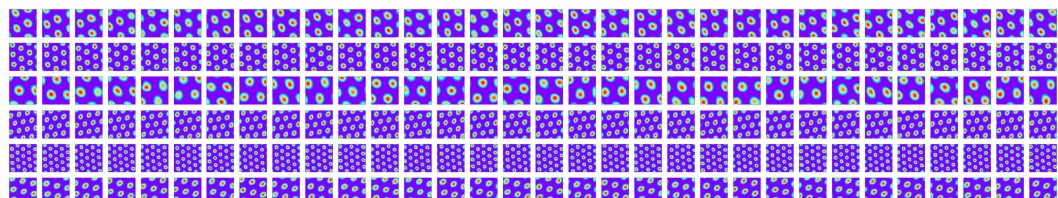

Figure 7: Learned patterns of $\boldsymbol{u}(\boldsymbol{x})$ with 6 blocks of size 32 cells in each block. Every row shows the learned patterns within the same block.

We further evaluate the spatial profile of the learned patterns with 6 blocks of size 16 using the same measures as in the main text. All learned patterns exhibit significant hexagonal periodicity in terms of gridness scores (mean 1.06, std 0.27, range 0.58 to 1.48), which exceed the 95 percentile of null distributions obtained by applying spatial field shuffle to each response map. The grid scales of learned patterns (mean 0.38, range 0.27 to 0.56), as shown in figure 10a, follows a multi-modal distribution. The ratio between neighbouring modes are roughly 1.37 and 1.38. As shown in figure 10b, the grid orientations of learned patterns are also multi-modal distributed.

Figure 11 shows the learned patterns of a block of $\boldsymbol{B}(\theta)$ over $\theta$ from 0 to $2\pi$. Regular sine/cosine tunings emerge. $(\boldsymbol{B}(\theta))$ can be isometric to $(\theta)$, in the sense that for each column $i$, the angle between $\boldsymbol{B}_i(\theta)$ and $\boldsymbol{B}_i(\theta + \Delta\theta)$ is $\Delta\theta$.

## B.3    SPATIAL PROFILE OF LEARNED HEXAGON GRID PATTERNS

Figure 12 shows the spatial profile of the patterns of $\boldsymbol{v}(\boldsymbol{x})$ over the 80 lattice.

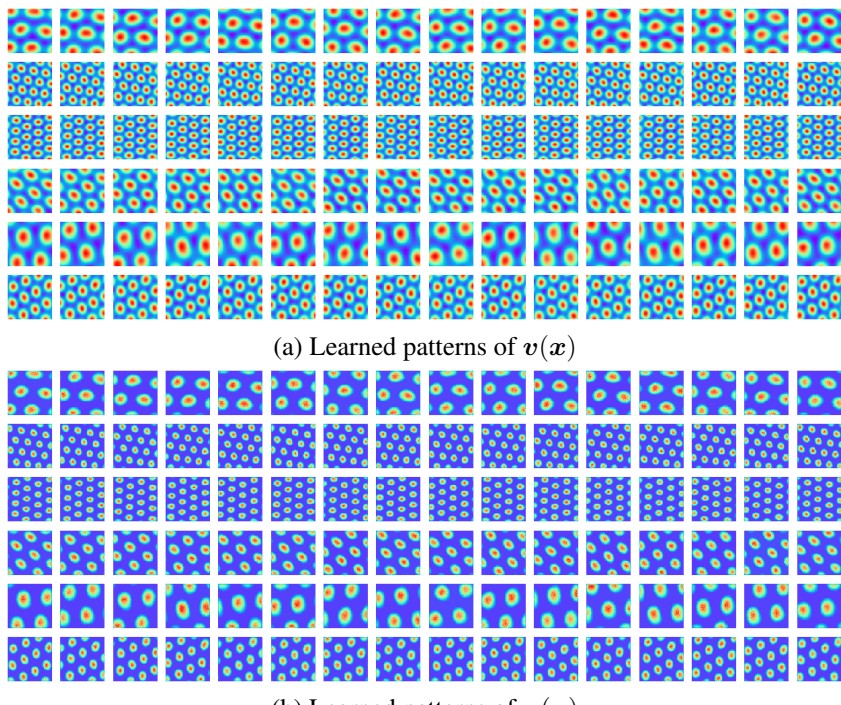

(a) Learned patterns of $\boldsymbol{v}(\boldsymbol{x})$

(b) Learned patterns of $\boldsymbol{u}(\boldsymbol{x})$

Figure 8: Learned patterns with 6 blocks of size 16. *Top*: $\boldsymbol{v}(\boldsymbol{x})$. *Bottom*: $\boldsymbol{u}(\boldsymbol{x})$. Every row shows the learned patterns within the same block.

### B.4 LEARNED PATTERNS IN ABLATION STUDY

Figure 13 shows the learned patterns of $\boldsymbol{v}(\boldsymbol{x})$ where the model is trained with a certain component removed.

## C PATH PLANNING

Define and model the adjacency according to Stachenfeld et al. (2017),

$$A_\gamma(\boldsymbol{x}, \boldsymbol{x}') = \mathbb{E}\left[\sum_{t=0}^{\infty} \gamma^t 1(\boldsymbol{x}_t = \boldsymbol{x}') | \boldsymbol{x}_0 = \boldsymbol{x}\right] = \langle \boldsymbol{v}(\boldsymbol{x}), \boldsymbol{u}_\gamma(\boldsymbol{x}') \rangle, \tag{29}$$

where $\mathbb{E}$ is with respect to a random walk exploration policy, and $\gamma$ is the discount factor that controls the temporal and spatial scales. We can discretize $\gamma$ into a finite list of scales. The above model, i.e., $\boldsymbol{v}(\boldsymbol{x})$ and $\boldsymbol{u}_\gamma(\boldsymbol{x}')$, can be learned by temporal difference learning. The basis expansion model with $d \ll N^2$ enables efficient learning from small amount of explorations, so that we can fill in unexplored $A_\gamma(\boldsymbol{x}, \boldsymbol{x}')$ based on the learned $\boldsymbol{v}(\boldsymbol{x})$ and $\boldsymbol{u}_\gamma(\boldsymbol{x}')$.

For random walk diffusion in open field, $A_\gamma(\boldsymbol{x}, \boldsymbol{x}') \propto \exp(-|\boldsymbol{x} - \boldsymbol{x}'|^2/2\sigma_\gamma^2)$, where $\sigma_\gamma^2$ depends on $\gamma$.

For random walk in a field with obstacles or non-Euclidean geometry, using Varadhan's formula (Varadhan, 1967), $A_\gamma(\boldsymbol{x}, \boldsymbol{x}')$ can still be approximated by Gaussian kernel except $|\boldsymbol{x} - \boldsymbol{x}'|$ is replaced by geodesic distance.

After learning $\boldsymbol{v}(\boldsymbol{x})$, $\boldsymbol{u}_\gamma(\boldsymbol{x}')$, we propose to use the following method for path planning. Let $\hat{\boldsymbol{x}}$ be the target. Let $\boldsymbol{x}^{(t)}$ be the current position, encoded by $\boldsymbol{v}(\boldsymbol{x}^{(t)})$, we propose to plan the next displacement by

$$\Delta\boldsymbol{x}^{(t+1)} = \arg\max_{\Delta\boldsymbol{x}} \langle \boldsymbol{M}(\Delta\boldsymbol{x})\boldsymbol{v}(\boldsymbol{x}^{(t)}), \boldsymbol{u}_\gamma(\hat{\boldsymbol{x}}) \rangle, \tag{30}$$

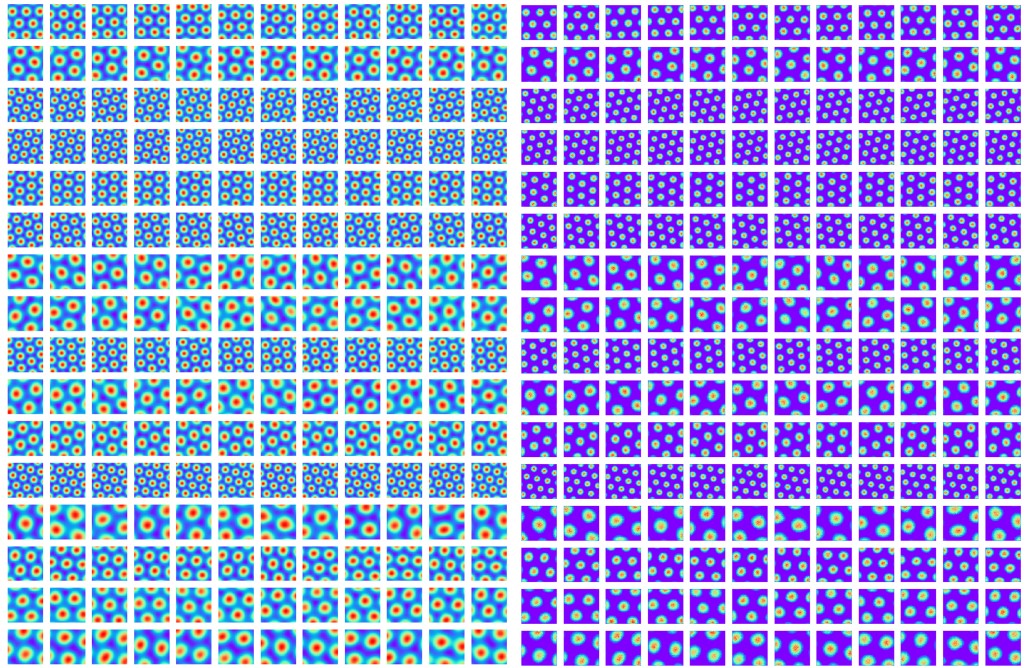

Figure 9: Learned patterns with 16 blocks of size 12. *Left*: $\boldsymbol{v}(\boldsymbol{x})$. *Right*: $\boldsymbol{u}(\boldsymbol{x})$. Every row shows the learned patterns within the same block.

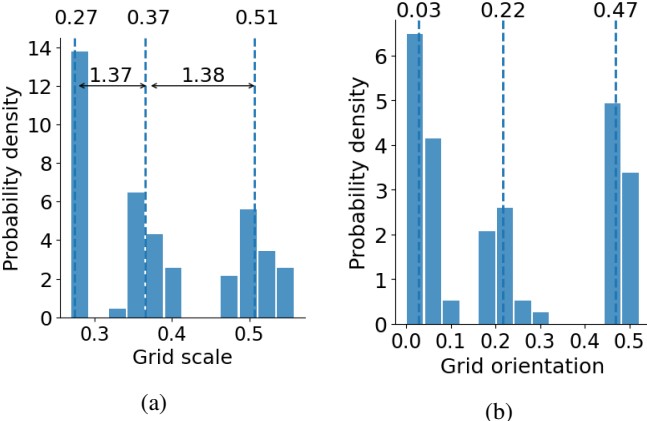

Figure 10: (a) Multi-modal distribution of grid scales. (b) Multi-modal distribution of grid orientations.

and let $\boldsymbol{x}^{(t+1)} = \boldsymbol{x}^{(t)} + \Delta\boldsymbol{x}^{(t+1)}$, encoded by $\boldsymbol{v}(\boldsymbol{x}^{(t+1)}) = \boldsymbol{M}(\Delta\boldsymbol{x}^{(t+1)})\boldsymbol{v}(\boldsymbol{x}^{(t)})$. In the above maximization, $\Delta\boldsymbol{x}$ is chosen from all the allowed displacements for a single step, and we also need to select an optimal $\gamma$ that is most sensitive to the change of $A$. An example of scale selection scheme is that we choose the smallest $\sigma$ that satisfies $\max_{\Delta\boldsymbol{x}}\langle \boldsymbol{M}(\Delta\boldsymbol{x})\boldsymbol{v}(\boldsymbol{x}^{(t)}), \boldsymbol{u}_\gamma(\hat{\boldsymbol{x}})\rangle > 0.2$. When $\boldsymbol{x}^{(t)}$ is far from $\hat{\boldsymbol{x}}$, the selected $\sigma_\gamma^2$ is big. When $\boldsymbol{x}^{(t)}$ is close to $\hat{\boldsymbol{x}}$, the selected $\sigma_\gamma^2$ is small. The above method enables automatic selection of scale. We shall explore other schemes of selecting $\gamma$ in future work.

We test path planning in open field using the learned model. Specifically, the model is first learned using a single scale $A_\gamma(\boldsymbol{x}, \boldsymbol{x}')$, where $\sigma_\gamma = 0.07$. Then we assume a list of three scales of $A_\gamma(\boldsymbol{x}, \boldsymbol{x}')$, i.e., $\sigma_\gamma = [0.07, 0.14, 0.28]$, and learn three corresponding sets of $\boldsymbol{u}_\gamma(\boldsymbol{x}')$. For planning, we create a pool of allowed displacements from which $\Delta\boldsymbol{x}$ is chosen: the length of $\Delta\boldsymbol{x}$ can be 1 or 2 grids, and the direction can be chosen from 200 discretized angles over $[0, 2\pi]$. Figure 14 depicts several

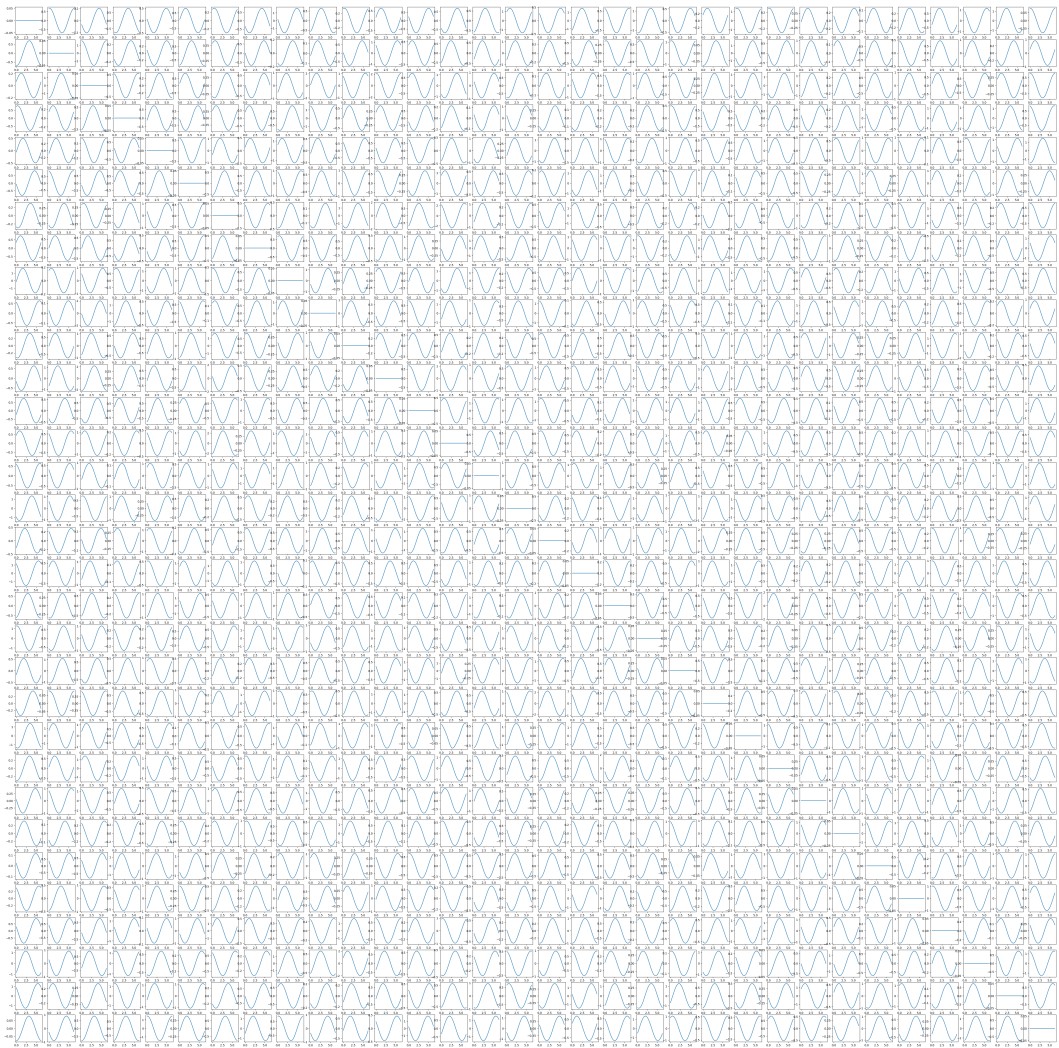

Figure 11: Learned patterns of a block of $\boldsymbol{B}(\theta)$. Each curve shows the patterns of one element of $\boldsymbol{B}(\theta)$ over $\theta \in [0, 2\pi]$. Since $\boldsymbol{B}(\theta)$ is skew-symmetric, the diagonal values are zeros and the value of $B_{ij}(\theta)$ is the same as the value of $B_{ji}(\theta)$. Regular sine/cosine tunings emerge.

examples of planned paths. As the examples show, when $\boldsymbol{x}^{(t)}$ is far from the target, kernel with large $\sigma_\gamma$ is chosen, and as $\boldsymbol{x}^{(t)}$ approaches the target, kernel with smaller $\sigma_\gamma$ is chosen. A planning episode is treated as successful if the distance between $\boldsymbol{x}^{(t)}$ and target is smaller than 0.5 grid within 40 time steps. In the cases where the distance between the starting point of the agent and the target is smaller than 20 grids, the successful rate is $100\%$ (test for $10,000$ episodes).

We shall explore this method in irregular fields with obstacles in future work.

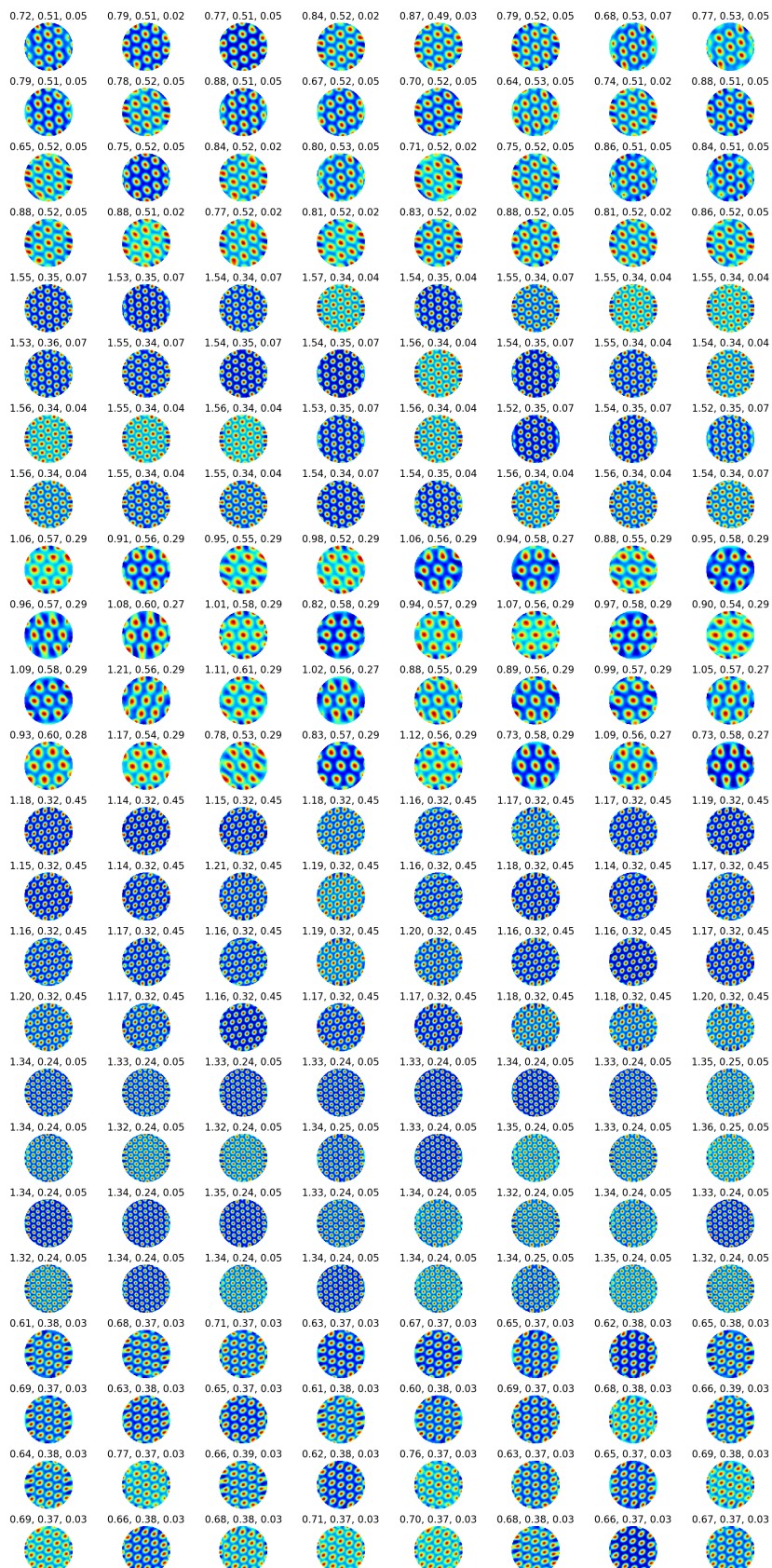

Figure 12: Spatial profile of the patterns of $v(x)$ over the $80 \times 80$ lattice. For each unit, the autocorrelogram is visualized. Gridness score, scale and orientation are listed sequentially on top of the autocorrelogram.

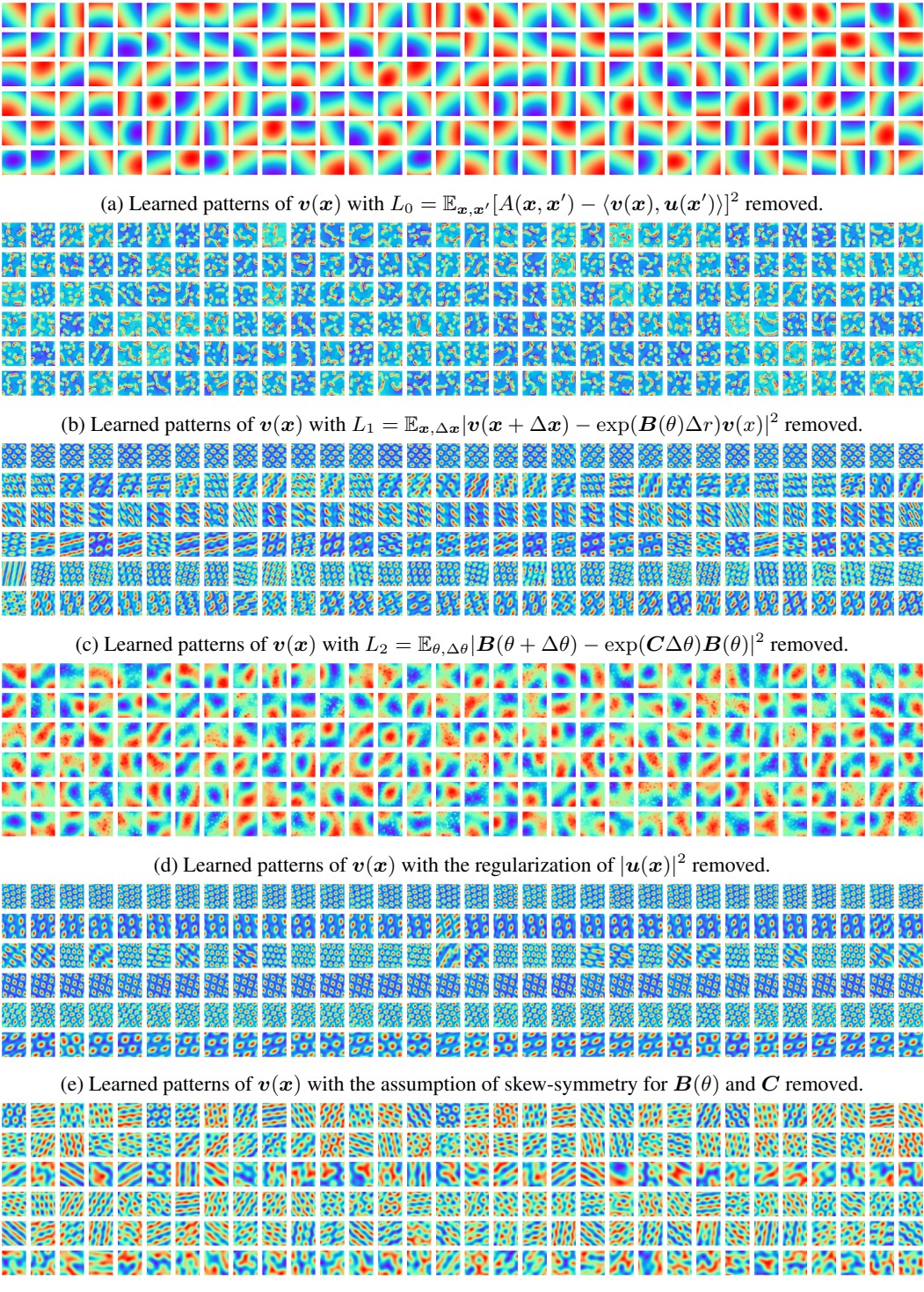

(a) Learned patterns of $v(x)$ with $L_0 = \mathbb{E}_{x,x'}[A(x,x') - \langle v(x), u(x')\rangle]^2$ removed.

(b) Learned patterns of $v(x)$ with $L_1 = \mathbb{E}_{x,\Delta x}|v(x+\Delta x) - \exp(B(\theta)\Delta r)v(x)|^2$ removed.

(c) Learned patterns of $v(x)$ with $L_2 = \mathbb{E}_{\theta,\Delta \theta}|B(\theta+\Delta\theta) - \exp(C\Delta\theta)B(\theta)|^2$ removed.

(d) Learned patterns of $v(x)$ with the regularization of $|u(x)|^2$ removed.

(e) Learned patterns of $v(x)$ with the assumption of skew-symmetry for $B(\theta)$ and $C$ removed.

(f) Learned patterns of $v(x)$ with the assumption of $u > 0$ removed.

Figure 13: Learned patterns of $v(x)$ in ablation study, where the model is trained with a certain component removed.

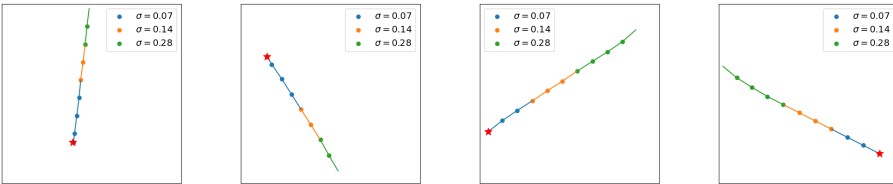

Figure 14: Examples of path planning in open field. Red star denotes the target. Three scales of $A_\gamma(\boldsymbol{x}, \boldsymbol{x}')$ are used ($\sigma_\gamma = [0.07, 0.14, 0.28]$). Different colors denote the kernel chosen at each step.

