# OpenReview forum: "A Representational Model of Grid Cells' Path Integration Based on Matrix Lie Algebras"
_ICLR.cc/2021/Conference — Reject_

### Official Review · AnonReviewer2 · 2020-10-27
**Elegant formalism but unclear utility**

**Rating:** 5
**Confidence:** 5

**Review:**

This paper proposes a simple recurrent model of how grid cells may perform path integration, which also recapitulates the well-known finding that grid cells exhibit hexagonal firing patterns. Their model consists of two primary components where self-position is represented by a population activity vector (which is rotated by a generator matrix of a Lie algebra whenever the agent moves in a given direction), and where self-motion is represented by the rotation of this vector (whereby when the agent changes its direction, this generator matrix is itself rotated by another generator matrix).

Strengths:
+ I think the model is a mathematically elegant formalism.
+It is more explicit than the approach taken by task-optimized (nonlinear) RNNs.

Weaknesses:
-	It is unclear to me what scientific insight we get from this model and formalism over the prior task-optimized approaches. For instance, this model (as formulated in Section 2.3) is not shown to be a prototype approximation to these non-linear RNN models that exhibit emergent behavior. So it is not clear that your work provides any further “explanation” as to how these nonlinear models attain such solutions purely through optimization on a task.
-	Furthermore, I am not really sure how “emergent” the hexagonal grid patterns really are in this model. Given partitioning of the generator matrices into blocks in Section 2.5, it almost seems by construction we would get hexagonal grid patterns and it would be very hard for the model to learn anything different.

While the ideas of this paper are mathematically elegant, I do not see the added utility these models provide over prior approaches nor how they provide a deeper explanation of the surprising emergent grid firing patterns observed in task-optimized nonlinear RNNs. For these reasons, I recommend rejection.

---

> ### Author Response · Authors · 2020-11-22
> **Thank you for your helpful feedback (1/2)**
>
> Thank you for your comments. We appreciate your judgement that our formalism is mathematically elegant and is more explicit than task-optimized RNNs.
>
> - Q1: About non-linear RNNs.
>
> A1: For infinitesimal displacement $\delta r$, our RNN follows the Lie algebra: $v(x+\delta x) = (I + B \delta r) v(x)$, which is linear in both $\delta r$ and $v(x)$. For finite (non-infinitesimal) $\Delta r$, the RNN follows the Lie group: $v(x+\Delta x) = \exp(B \Delta r) v(x)$, where $\exp(B \Delta r)$ forms a Lie group and takes the form of exponential map, which is derived from the Lie algebra. This RNN is linear in $v(x)$, but it is non-linear in $\Delta r$. Thus our RNN model does explain some non-linearity in RNN.
>
> We would like to emphasize that our main goal is to construct a minimal model of the path integration process while accounting for the response properties of neurons in entorhinal cortex. Our results suggest that a simple approach without relying on complicated non-linearity is sufficient to do so. It would be interesting for future work to incorporate more general non-linearity to our current model to further understand the relation between such models and previous non-linear RNN models.
>
> - Q2: About how emergent the hexagonal grid patterns are.
>
> A2: Our model, i.e., equations (1) and (2), is based on generic vector and matrices, and is almost minimally simple. We did not make any specific assumptions about hexagonal grid patterns, such as specific form of Fourier expansion. Crucially, these patterns are not a simple or trivial consequence of the block-diagonal structure of the connectivity. This could also be seen in Section 5.3 where we systemically removed components in our model and investigated the effects on the firing patterns. Hexagon patterns do not emerge if certain component is removed, even if the block-diagonal structure is still applied. The fact that such a simple model can learn such clean hexagon grid patterns is a strong indication that our model may provide a deep understanding of the grid cells.
>
> Section 2.2 presents some analysis of our model. We added a new Section 2.3 in the revised version to connect Proposition 1 to the hexagonal grid patterns. The fact that hexagonal grid patterns can emerge from equations (1) and (2) seems non-trivial.

---

> > ### Author Response · Authors · 2020-11-22
> > **Thank you for your helpful feedback (2/2)**
> >
> > - Q3: About scientific insight and added utility.
> >
> > A3: (1) We propose a simple and explicit representational model for the path integration calculation of grid cells, where the 2D self-position is represented by a vector in neural space, and 2D self-motion is explicitly represented by the rotation of vector. Despite the simplicity of our model, our model can explain key neuroscience observations of grid cells. Our model differs from previous studies in that our formulation explicitly represents the continuous self-motion.
> >
> > Proposition 1 in Section 2.2 says that if the agent moves by $\delta r$, then the vector rotates by an angle $\beta \delta r$. By the same argument, for each module $k$, the sub-vector $v_k(x)$ rotates by an angle $\beta_k \delta r$. The parameter $\beta_k$ tells us how fast the sub-vector rotates as the agent moves. It explicitly captures the metric of each module. Such an explicit model may provide a deeper understanding of grid cells.
> >
> > Under our model, for the hexagonal grids to emerge, we do not require the firing fields of the place cells to exhibit a Mexican-hat (balanced excitatory center + inhibitory surround). This is an assumption made in several previous papers (Dordek et al, eLife, 2016; Sorscher et al NeurIPS 2019), but nonetheless with no experimental evidence so far. In contrast, our model simply requires experimentally observed Gaussian-like place fields. We consider this as an advantage over the previous studies. It suggests that by incorporating the constraints that is generic to the path integration (i.e., symmetry in translation) as we did in our model, the hexagonal grid pattern could be naturally explained without invoking additional assumptions such as inhibition surround of place fields that have not received experimental support so far.
> >
> > (2) In terms of general representation learning, the key contribution of our work is about representing continuous motion, or continuous transformation, or continuous relation. While it is now a common practice to represent state by vector in neural space, the representation of continuous motion has not received an in-depth treatment so far. The continuous motion in physical space usually has a native Lie algebra and Lie group structure. Our work represents them by matrix Lie algebra and matrix Lie group that act in neural space. We believe our method is generally applicable to modeling the head direction system, as well as modeling the motor cortex for representing the continuous motions of arm, hand, pose, etc.
> >
> > (3) Comparison with general RNN. The general RNN in the literature does not represent continuous motion or continuous transformation explicitly, much less represent the native Lie algebra and Lie group structure of the continuous motion in physical space. In contrast, our model provides explicit representation of continuous transformation and its associated algebraic structure.
> >
> > To be more specific, consider an RNN: $v(x+\Delta x) = {\rm RNN}(v(x), \Delta x)$, for a finite (non-infinitesimal) $\Delta x$. This RNN transformation should be the same as composing a sequence of $N$ steps of RNN transformations whose inputs are $\Delta x/N$, even as $N$ goes to infinity. Our work provides a simple form of RNN that satisfies this constraint. It appears that our simple RNN can already explain the path integration and hexagon patterns of grid cells.
> >
> > In science, we always seek the simplest model for explanation. Given the simplicity and elegance (thanks for your comment in this regard) of our model and the advantage of our model over the previous ones, we humbly wish you could reconsider your evaluation of our paper. Thank you.

---

> ### Author Response · Authors · 2020-11-24
> **One more point**
>
> As for the model prototype, the basis expansion model (Dordek et al, eLife, 2016; Sorscher et al NeurIPS 2019) is linear in $v(x)$. It is perhaps desirable to have path integration model to be linear in $v(x)$ too. The rotation of $v(x)$ is capable of path integration. The way $v(x)$ rotates can explain hexagonal periodic patterns and metric of module.
>
> It can be satisfying to have both models linear in $v(x)$ to explain path integration and the interaction between grid cells and place cells. The connection between the two models is also mathematically based on group representation theory.

---

### Official Review · AnonReviewer3 · 2020-10-28
**Path Integration Lie Algebra**

**Rating:** 8
**Confidence:** 5

**Review:**

Summary: The authors propose a simple recurrent network as a model of spatial navigation in the MEC/Hippocampal network. This model assumes that grid cells only regularly receive egocentric movement information, an important aspect for understanding the origin of these functional cell types in-vitro. Overall, this article should be of interest for any ICLR members interested in biological  spatial navigation.

Strong Points:
As stated above, the model is an intuitive explanation of how allocentric-egocentric transformations might be performed. While the authors use a back propagation approach to training, the separated loss functions (eq 10-12) for each layer of the network mean that learning could be performed by predictive contrastive coding, as recent research suggests biological networks may be doing.
The learned receptive fields show many of the more nuanced aspects of grid cells found in experimental studies, such as discretized angle relative to the environment. The investigation of error accumulation as a function of time steps since encoding allows additional comparisons to the literature. These give additional confidence that the model is biologically plausible, at some level of abstraction.

Weak Points: As a non-mathematician, it's unclear to me what the implications of the lie algebra presented in the beginning of section 2 are. My understanding is that this makes grid activities the product of two separable matrices (displacement, and rotation). While this is biologically plausible, perhaps the authors could explain any additional reasons why this is of importance.

Additional Comments: A possibly interesting future experiment would be investigating effects on error correction by only having place cells at a handful of spatial scales (A.2), especially if there is a "block" structure in u.

---

> ### Author Response · Authors · 2020-11-22
> **Thank you for your helpful feedback**
>
> We deeply appreciate your evaluation of our work, including its appropriateness for the ICLR community.
>
> Thank you for your comments on predictive contrastive coding. We shall study it in our future work. We shall also study the effect on error correction by having place cells at a small number of spatial scales in our future work.
>
> About mathematical implications of the Lie algebra. You are right that our model couples two rotation systems driven by two sets of generator matrices, one for displacement, and the other for the change of direction. Along each direction, the displacement is represented by rotation driven by a generator matrix. The generator matrices at different directions themselves are rotated copies of each other. This enables us to embed the polar coordinate system at x into the neural space of $v(x)$, as illustrated by Figure 1.
>
> Proposition 1 in Section 2.2 elucidates the implication of our model. If the agent moves by $\delta r$, the vector rotates by an angle $\delta \alpha = \beta \delta r$, where $\beta$ is the same for all the directions of self-motion. $\beta$ tells us how fast the vector rotates as the agent moves. For the learned model, $\beta$ can be much bigger than 1, so that the vector rotates back to itself in a short distance. This leads to periodic patterns of grid cells. The rotation is locally isotropic in that $\beta$ is the same for all directions. This may underly the hexagon grid patterns.
>
> Specifically, the hexagon grid patterns can be created by linearly mixing three Fourier plane waves whose directions are $2 \pi/3$ apart. According to Gao et al. ICLR 2019 cited in the paper, if $v(x)$ is an orthogonal mixing of three such Fourier plane waves, then the angle between $v(x)$ and $v(x+\delta x) = \beta \delta r$, where $\delta r$ is the length of $\delta x$, and $\beta$ is the same for all the directions, i.e., isotropic. This is exactly the property we derive in Proposition 1.
>
> We have added a new Section 2.3 to explain the implications of our Lie algebra model. We are pursuing a more general analysis of our model.

---

### Official Review · AnonReviewer4 · 2020-10-30
**Interesting suggestion backed by numerical experiments but lacking analytical justification**

**Rating:** 5
**Confidence:** 4

**Review:**

The authors develop a model for learning the observed responses of grid cells (GC) in the entorhinal cortex from the animal movement vectors. Their key assumption is that the GC activity vector rotates with the movement magnitude according to the Lie group formalism and the corresponding Lie group generator is also rotated by the change in movement orientation. Their claim is supported by a numerical optimization of the objective function reflecting these assumptions as well as the projection onto the place cell representation in the hippocampus.

I find the paper original, interesting and clearly written. The numerical simulations support the claim. However, given the current state of the field, I would like to see an analytical demonstration of this claim like in the work of Sorscher et al which the authors cite. Specifically, would it be possible to demonstrate that dropping the orthogonality constraint used in Sorscher et al and introducing the rotation of the Lie generator would still result in realistic GC responses as the authors claim? Such analytical demonstration would provide a much needed insight into the operation of the system.

Minor comments:

Page 3, second line after Eq(4): the closing parenthesis is missing after B(\theta

Sorscher et al reference is listed twice

%%%%%%%%%%%%%%%%%%%%%%%%%%%

Added after author response. My enthusiasm for the paper has diminished because it seems to be more of an incremental step over Gao et al 2019 and the authors did not provide additional analytical insight into their new results.

---

> ### Author Response · Authors · 2020-11-22
> **Thank you for your helpful feedback**
>
> Thank you for the very insightful summary of our work. We reply to your comments in order.
>
> - Q1: Comparison with  Sorscher et al. NeuIPS 2019. Would it be possible to demonstrate that dropping the orthogonality constraint used in Sorscher et al and introducing the rotation of the Lie generator would still result in realistic GC responses as the authors claim?
>
>
> A1: Yes. The relationship between our model and the work of Sorscher et al. is that we keep the basis expansion model as in Sorscher et al., i.e., the loss term $L_0$, but we drop the orthogonality constraint in Sorcher et al., and replace the orthogonality constraint by our vector rotation model drive by Lie generators, i.e., the loss terms $L_1$ and $L_2$, corresponding to equations (1) and (2) of our model. Our model is able to learn realistic GC responses without orthogonality constraint.
>
> Replacing the orthogonality constraint by our vector rotation model enables our model to perform path integration calculation.
>
> Importantly, for the hexagonal grids to emerge in our model, we do not require the firing fields of the place cells to exhibit a Mexican-hat (balanced excitatory center + inhibitory surround), a critical assumption made in several previous papers (Dordek et al, eLife, 2016; Sorscher et al NeurIPS 2019), but nonetheless with no experimental evidence so far. Our model simply requires Gaussian place fields, which are in nice agreement with the experimental data. This difference also suggests the advantage of our approach.
>
> In our model, we assume grid cells form modules or groups, so that $v(x)$ consists of sub-vectors $(v_k(x), k = 1, …, K)$. The learned response maps of grid cells in different modules tend to be orthogonal to each other. But the response maps of grid cells in the same module are not orthogonal to each other. See Section A.3. It is possible that each module may consist of many grid cells, and it may not be biologically realistic to assume that their response maps are orthogonal to each other.
>
>
> - Q2: Analytical demonstrations.
>
> A2: We agree with you that it is important to have analytical demonstrations.
>
> Proposition 1 provides key analysis of our model. It says that when the agent moves by $\delta r$, the vector rotates by an angle $\delta \alpha$, and $\delta \alpha = \beta \delta r$, where $\beta$ is the same for all the directions of self-motion, i.e., isotropic.
>
> $\beta$ tells us how fast the vector rotates as the agent moves. For the learned model, $\beta$ is much larger than 1, so that as the agent moves, the vector quickly rotates back to itself in a short distance. This causes the periodic patterns of the response maps of the grid cells.
>
> Because $\beta$ is the same for all the directions of self-motion, the rotation is isotropic for the same $\delta r$. This isotropy may underline the hexagon grid patterns.
>
> More specifically, the hexagon grid patterns can be obtained by linearly mixing Fourier plane waves whose directions are $2 \pi/3$ apart. In the revised version of our paper, we adapted a result in Gao et al. ICLR 2019 as Proposition 2, which states that if $v(x)$ is an orthogonal mixing of three Fourier plane waves whose directions are $2 \pi/3$ apart, then the angle between $v(x)$ and $v(x+\delta x)$ is $\beta \delta r$, where $\delta r$ is the length of $\delta x$, and $\beta$ is the same for all the directions of $\delta x$. Proposition 1 proves that this property emerges from our model.
>
> We have added a new Section 2.3 to connect Proposition 1 to hexagon grid patterns. We are pursuing a more general analysis of equations (1) and (2) of our model.
>
>
> - Q3: Minor comments.
>
> A3: We have fixed them in our revision. Thank you.

---

### Official Review · AnonReviewer1 · 2020-10-31
**An interesting approach but not clear what we learn from the result**

**Rating:** 6
**Confidence:** 4

**Review:**

The paper proposes a model of the grid cell system based on computing an embedding of position and head direction that allows matrix Lie algebras to translate and rotate the coordinate frame.  The results show that individual elements of the embedding have spatial response profiles resembling grid cells in entorhinal cortex.

Equations 1 and 2 are a nice mathematical approach.  the justification for using Lie algebra is clear and its very elegant, and seems like a good idea.   The results are interesting too - very neat to see the hexagonal grid arrangement emerge.  But why?  Here I'm not sure we learn anything.  In fact the authors seem to state that certain parameter settings such as beta are required to bring about this result, but there's little insight provided as to why.  And how does it affect the result?  Also, the modularity - one of the most interesting facets of the grid cells system - is rigged in advance, rather than an emergent property.  So it leaves you wondering, what's the point here?  For example, what is the advantage conferred by embedding 2D position into a higher dimensional space?  why is this a good thing for the brain to do?  and why a grid system?  the answer seems to be "that's what emerges from our matrix Lie algebra model."  ok, but that's not very illuminating.  Surely there should be some way to reason about this result and why its useful for brains to have this type of representation.  That would make the paper a lot more interesting in my view.

The authors do a nice job investigating the influence of different terms.  Ok, but it also seems like just hunting and pecking phenomenology, try this try that.

Also this sentence in the intro I find rather unsatisfying:  "It is worth noting that our work is mainly concerned with representation learning. We do not seek to pursue biologically realistic modeling of neural dynamics.." - so you are modeling an aspect of biology, but you aren't concerned with realistic modeling of neural dynamics?  I can understand perhaps leaving out spiking neurons and ion channels and all that, but it would seem to me that taking into account what is actually neuronally feasible is important.  For example, presumably having a local connectivity structure in B is important, and that could well affect the results.  Its not just about representation learning, but implementation and biophysical constraints are also important.

---

> ### Author Response · Authors · 2020-11-22
> **Thank you for your helpful feedback (1/2)**
>
> Thank you for your insightful comments and questions. The following are our replies to your points.
>
> - Q1: About our model.
>
> A1: Our work seeks to understand how the brain may perform path integration calculation. We propose a simple and novel representational model: we represent the 2D self-position by a vector in a high-dimensional neural space, and represent the self-motion by the rotation of the vector, which can be mathematically formulated by matrix Lie algebra and Lie group. We demonstrated that the learned model is capable of accurate path integration. The learned response maps of model neurons replicate several key neuroscience observations about grid cells. Thus, our work contributes a new understanding of grid cells. It also provides a new class of models for representing continuous motion or transformation.
>
>
> - Q2: About the $\beta$ parameter in Proposition 1.
>
> A2: This parameter is a consequence of our model. Specifically, we prove that if the agent moves by $\delta r$, then the vector rotates by an angle
>
> $ \delta \alpha = \beta \delta r$.  (*)
>
> Such a setting emerges from our model, i.e., equations (1) and (2). We do not make any explicit assumption about $\beta$.
>
>
> - Q3: Theoretical understanding of our model and hexagon grid patterns.
>
> A3: Equation (*) is a key property of our model. It says that when the agent moves by $\delta r$, the vector rotates by an angle $\beta \delta r$, and $\beta$ is isotropic in that it is the same for all directions of self-motion. $\beta$ tells us how fast the vector rotates as the agent moves. For the learned model, in practice $\beta$ can be much bigger than 1, so that the vector will quickly rotate back to itself after a short distance, causing the periodic pattern. The local isotropic property of $\beta$ appears to further cause the periodic pattern to be hexagonal.
>
> Specifically, the hexagon grid patterns can be obtained by linear mixing of three Fourier plane waves whose directions are $2 \pi/3$ apart. To connect Proposition 1 to hexagon grid patterns, we adapted a theoretical result from Gao et al, ICLR 2019 [1] as Proposition 2 in the revised version, which says that if the vector $v(x)$ is an orthogonal mixing of three Fourier plane waves that are $2 \pi/3$ apart, then the angle between $v(x)$ and $v(x+\delta x)$ is $\beta \delta r$, where $\delta r$ is the length of $\delta x$. That is, such $v(x)$ satisfies the property (*) above.
>
> We have added a new subsection, Section 2.3, to connect Propositions 1 to hexagon grid patterns. We are currently pursuing a more general theoretical analysis of equations (1) and (2).
>
>
> - Q4: About modularity.
>
> A4: We do assume that the generator matrices are block diagonal, where each block corresponds to a module, so that the vector consists of sub-vectors, each of which is rotated by a sub-matrix.  By the same argument as in the proof of Proposition 1, when the agent moves by $\delta r$, each sub-vector $v_k(x)$ rotates by an angle $\delta \alpha_k = \beta_k \delta r$. The parameter $\beta_k$ explicitly captures the scale or metric of each module $k$.
>
> In the context of our model, the modularity assumption is to assume that the rotations of the sub-vectors are disentangled, i.e., each sub-vector is driven by a separate recurrent network. This assumption appears to be biologically plausible. We believe modularity is part of the design of the recurrent network.
>
>
> - Q5: Advantage of embedding 2D position into a higher dimensional space.
>
> A5: Embedding 2D position x into a higher dimensional neural space as $v(x)$ is not unique to our model. In fact, it is fairly standard in computational neuroscience to use high-dimensional neural population codes to represent low-dimensional variables. For example, it is also assumed in the pioneering papers on RNN models for path integration [2][3], where $v$ is driven by an RNN. It is also assumed in the pioneering papers on basis expansion model [4][5], where $v(x)$ serves as basis functions for expanding place cells, i.e., the Gaussian kernel $A_{x’}(x) = \langle v(x), u(x’)\rangle$.
>
> One advantage of using $v(x)$ is that our simple recurrent model that is linear in $v(x)$ can represent self-motion for path integration. Another advantage is error correction (see Section 5.2).
>
> Compared to previous papers mentioned above, we make the embedding idea more explicit. In addition to representing 2D self-position by a vector, we also represent 2D self-motion by the rotation of the vector. Representing continuous motion or continuous transformation is a key contribution of our paper.

---

> > ### Author Response · Authors · 2020-11-22
> > **Thank you for your helpful feedback (2/2)**
> >
> > - Q6: Realistic modeling of neural dynamics.
> >
> > A6: While we do not seek to model detailed realistic neural dynamics, we agree with you that implementation and biophysical constraints are important aspects of the grid cell systems. Our model is similar to the pioneering papers on path integration [2][3] and basis expansion model [4][5], in terms of biological realism. We have removed the controversial sentence about biological realism. Thanks.
> >
> >
> > References:
> >
> > [1] Ruiqi Gao, Jianwen Xie, Song-Chun Zhu, and Ying Nian Wu. Learning grid cells as vector rep- resentation of self-position coupled with matrix representation of self-motion. ICLR, 2019.
> >
> > [2] Andrea Banino, Caswell Barry, Benigno Uria, Charles Blundell, Timothy Lillicrap, Piotr Mirowski, Alexander Pritzel, Martin J Chadwick, Thomas Degris, Joseph Modayil, et al. Vector-based navigation using grid-like representations in artificial agents. Nature, 557(7705):429, 2018.
> >
> > [3] Christopher J Cueva and Xue-Xin Wei. Emergence of grid-like representations by training recurrent neural networks to perform spatial localization. ICLR, 2018.
> >
> > [4] Yedidyah Dordek, Daniel Soudry, Ron Meir, and Dori Derdikman. Extracting grid cell characteristics from place cell inputs using non-negative principal component analysis. Elife, 5:e10094, 2016.
> >
> > [5] Kimberly L Stachenfeld, Matthew M Botvinick, and Samuel J Gershman. The hippocampus as a predictive map. Nature neuroscience, 20(11):1643, 2017.

---

### Decision · Program_Chairs · 2021-01-07
**Final Decision**

**Decision:**

Reject

**Comment:**

The authors propose a recurrent model of self-position, with a handcrafted expression of the rotational structure in terms of a matrix Lie group. As noted by the reviewers, this work strongly builds upon Gao et al (ICLR 2019). This really is mentioned too late and not prominently enough in the manuscript, and furthermore, the difference to this work is not clearly explored in the paper (there are just two sentences immediately prior to the conclusion and no experimental comparison). The reviewers pointed out that the phenomena observed here are handcrafted into the structure of the model, rather than being emergent. The reviewers raised concerns that it is not clear what conclusion to draw from this work.
For these reasons, I recommend rejection this stage.